# Exploiting Task Relationships in Continual Learning via Transferability-Aware Task Embeddings

**Yanru Wu[1]** **Jianning Wang[2]** **Xiangyu Chen[1]** **Enming Zhang[1]**

**Hanbing Liu[1]** **Yang Tan[2]** **Yang Li[1]\***

[1]*Shenzhen International Graduate School, Tsinghua University*[†]
[2]*Independent Researcher*

{wu-yr21, xy-c21, zem24, liuhb24}@mails.tsinghua.edu.cn
jianning.wang@outlook.com  tanyang1231@163.com
yangli@sz.tsinghua.edu.cn

## Abstract

Continual learning (CL) has been a critical topic in contemporary deep neural network applications, where higher levels of both forward and backward transfer are desirable for an effective CL performance. Existing CL strategies primarily focus on task models — either by regularizing model updates or by separating task-specific and shared components — while often overlooking the potential of leveraging inter-task relationships to enhance transfer. To address this gap, we propose a transferability-aware task embedding, termed H-embedding, and construct a hypernet framework under its guidance to learn task-conditioned model weights for CL tasks. Specifically, H-embedding is derived from an information theoretic measure of transferability and is designed to be online and easy to compute. Our method is also characterized by notable practicality, requiring only the storage of a low-dimensional task embedding per task and supporting efficient end-to-end training. Extensive evaluations on benchmarks including CIFAR-100, ImageNet-R, and DomainNet show that our framework performs prominently compared to various baseline and SOTA approaches, demonstrating strong potential in capturing and utilizing intrinsic task relationships. Our code is publicly available at `https://github.com/viki760/Hembedding_Guided_Hypernet`.

## 1 Introduction

Continual learning (CL), also referred to as incremental learning or life-long learning, has become an essential topic in the modern application of deep neural networks. In CL settings, a model is expected to sequentially learn a series of tasks (typically involving either category shifts or data distribution changes [Qu et al., 2021]) to progressively enhance its capabilities [Wang et al., 2024]. A key practical challenge in this process is catastrophic forgetting (CF) [Kirkpatrick et al., 2017], a phenomenon where learning a new task undermines the knowledge acquired from previous ones.

The mitigation of CF has long been a primary focus in CL research [Wang et al., 2024] due to its severe hindrance to the overall accumulation of model capacity. Existing approaches can be broadly categorized into three classes: rehearsal-based methods (e.g., ER [Robins, 1995], DER [Buzzega et al., 2020]) maintain a memory buffer of previous samples for later replay; regularization-based

---

\*Corresponding Author

[†]Yanru Wu, Xiangyu Chen, Enming Zhang, Hanbing Liu, and Yang Li are affiliated with the Shenzhen Key Laboratory of Ubiquitous Data Enabling, SIGS, Tsinghua.

methods (e.g., LwF [Li and Hoiem, 2017], SI [Zenke et al., 2017]) constrain parameter updates to protect prior knowledge; and architecture-based methods (e.g., PackNet [Mallya and Lazebnik, 2018], WSN [Kang et al., 2022]) allocate task-specific and task-sharing components of the model from different architecture levels.

However, strategies that explicitly alleviate forgetting often compromise the acquisition of new knowledge, prioritizing retention over adaptability. In contrast, an effective CL system should facilitate the intrinsic knowledge transfer across all tasks, improving future task performance via *forward transfer*, and preserving (or even enhancing) prior task performance via *backward transfer* [von Oswald et al., 2020]. Moreover, most existing works adopt a model-centric perspective[3], relying on posterior information obtained during training to manage cross-task interactions. Yet in CL settings, task sequences often exhibit certain degrees of noise and diversity, making such approaches sensitive to sample- and task-level variations. This can lead to potential negative transfer, reduced robustness, and limited scalability, particularly as the number or complexity of tasks increases.

These observations point to a fundamental question in continual learning: *Robust and positive knowledge transfers across tasks rely on capturing the underlying task relationships, but how do we efficiently learn and utilize such relations in CL settings?* Despite the valuable contributions of previous explorations [e.g. Jin and Kim, 2022, Riemer et al., 2018], they mostly remain constrained by a posterior model-centric paradigm with its associated drawbacks, leaving the prior exploitation of task relations largely underexplored. Based on this insight, we argue that the incorporation of a prior, task-relation-aware guidance can effectively leverage task relation information and help mitigate instability and performance degradation in CL, especially over long or challenging task sequences.

Therefore, in this work we explicitly bring focus to the prior exploitation of task relationships by introducing a statistically grounded, task-relation-aware embedding, and propose a CL framework under its guidance. Specifically, recognizing the shared goal between identifying inter-task relationships and the role of transferability metrics [Ding et al., 2024] in evaluating source-target task compatibility, we propose an online embedding scheme named H-embedding, which distills task-level transferability into a low-dimensional representation through optimization before task training. H-embedding can be learned efficiently without revisiting previous samples by maximizing the consistency between the Euclidean distances among embeddings and the H-score transferability [Bao et al., 2019] among corresponding tasks. To ensure alignment between embedding distances and transferability scores, we apply analytic hierarchy process (AHP) normalization, which significantly improves learning stability over long task sequences while maintaining efficiency. Built upon this embedding, we present a hypernetwork-based CL framework, where a task-conditioned hypernetwork generates task-specific model weights based on H-embedding modulated task embeddings.

In summary, with the aim of better understanding and utilization of the task relationships in CL, we propose in this work a novel H-embedding guided hypernet framework. Our framework is featured by: 1) efficient and reliable learning of task embedding based on the information theoretical foundation of H-score metric; 2) ease of practical use with end-to-end training and minimal additional storage beyond low-dimensional task embeddings; 3) the flexibility to serve as a plug-in module for generating and substituting specific model parts, such as LoRA layers, therefore compatible with various existing CL strategies and pretrained models; 4) a notable enhancement of CL in overall performance across different model backbones and benchmark datasets.

## 2 Related Works

### 2.1 Continual Learning Strategies

Up to now, there have been many studies dedicated to CL strategies, most of them involving the rehearsal of previous data to alleviate the knowledge degradation caused by CF [e.g. Robins, 1995, Buzzega et al., 2020]. However, growing privacy and data safety concerns have made this solution not always feasible, bringing increased attention to the rehearsal-free CL setting Smith et al. [2023b]. Most rehearsal-free methods rely on regularization—either in parameter space [e.g. Kirkpatrick et al., 2017, Zenke et al., 2017] or feature space [e.g. Li and Hoiem, 2017, Rebuffi et al., 2017]—by constraining model updates to preserve past knowledge. While effective in CF mitigation, these methods are often based on assuming task similarity and pose limitations to model adaptability.

---

[3]More discussion in Sec. 2.

Architecture-based approaches [Wang et al., 2024] offer an alternative by separating task-specific and shared components at various architectural levels [e.g. Mallya and Lazebnik, 2018, Wortsman et al., 2020, Jin and Kim, 2022]. With the rise of pretrained models, parameter efficient finetuning (PEFT) components like prompts [e.g. Wang et al., 2022b, Smith et al., 2023a, Wang et al., 2023] and LoRA modules [e.g. Liang and Li, 2024, Wu et al., 2025] are notably gaining popularity in this paradigm. Nevertheless, the allocation of model parts to different tasks usually comes with scaling problems with the growth of task numbers, potentially resulting in insufficient model capacity or excessive model size growth. Task transferability has recently been explored to enhance CL performance [Ermis et al., 2022]. Yet, the current method relies on storing both past models and selected samples, which is incompatible with the rehearsal-free setting. Developing strategies that effectively incorporate task transferability while remaining compatible with the online CL paradigm therefore remains an open question.

## 2.2 Transferability Metrics

Task transferability [Zamir et al., 2018] investigates the relationships between tasks and provides an effective method to evaluate and select source tasks in transfer learning. It also plays a crucial role in developing strategies for multi-task learning and meta-learning. For ease of use, previous studies have proposed metrics based on task models and data distributions for a quick estimation of transferability [Ding et al., 2024]. H-score [Bao et al., 2019, Ibrahim et al., 2022, Wu et al., 2024] uses an information-theoretic framework to evaluate transferability by solving a maximum correlation problem. NCE [Tran et al., 2019] employs conditional entropy to assess transferability and task difficulty. LEEP score [Nguyen et al., 2020, Agostinelli et al., 2022] offers a more generalized metric, defined by measuring the performance of a classifier developed from source model predictions when applied to the target task. LogME [You et al., 2021] assesses target task accuracy using a formulation integrating all possible linear classifiers derived from source model features. OTCE [Tan et al., 2021, 2024] combines optimal transport with conditional entropy to both estimate the domain and task difference between source and target. These metrics are mostly designed with differed assumptions and source accessibility, with their use applicable to different problem settings.

# 3 Preliminary

## 3.1 Mathematical Formulation

Consider a problem setting consisting of $M$ tasks $\{T_j\}_{j=1}^M$, the data of task $j$ is denoted by $D_j = (X^{(j)}, Y^{(j)})$, with input samples $X^{(j)} = \{x^{(j,i)}\}_{i=1}^{N_j}$ and output samples $Y^{(j)} = \{y^{(j,i)}\}_{i=1}^{N_j}$. Here, $N_j = |X^{(j)}| = |Y^{(j)}|$ denotes the sample size of the $j$-th task, and the attributes of sample data $x^{(j,i)}, y^{(j,i)}$ depends on the particular CL setting as well as the form of tasks. In CL, the $M$ tasks are learned sequentially during the training stage. To be specific, denoting a neural network model as $f(x, \Theta)$ (where $f$ represents the model function, $x$ represents the input data, and $\Theta$ represents the model weights) and the model weights acquired in task $j-1$ as $\Theta^{(j-1)}$, the goal of learning task $j$ is to derive a new set of weights $\Theta^{(j)}$ that not only achieves the optimal performance on task $j$, but also performs better or not significantly worse than $\Theta^{(j-1)}$ on tasks $T_1, \ldots, T_{j-1}$. For a rehearsal-free CL setting, the previous data $D_1, \ldots, D_{j-1}$ are not accessible during the training of the $j$-th task.

## 3.2 Hypernets

Hypernets [Ha et al., 2017], or hypernetworks, are specialized neural networks that generate weights of another neural network, i.e., the target network. Given a target model or component $f$, a hypernetwork $f_h$ produces its parameters $\Theta$ conditioned on an auxiliary input. This input can be naturally interpreted as a task embedding, as it reflects the task-specific information needed to generate appropriate parameters. Denoting the task embedding as $e$ and hypernet parameters as $\Theta_h$, we have $\Theta = f_h(e, \Theta_h)$. By leveraging task embeddings, hypernets enable flexible parameter modulation across tasks, offering a mechanism for task adaptation without direct weight sharing or parameter interference. Recently, hypernets have gained recognition as a potent tool in deep learning, proving effective across diverse deep learning tasks [Chauhan et al., 2023], including continual learning [von Oswald et al., 2020], causal inference [Chauhan et al., 2024], domain adaptation [Volk et al., 2022], few-shot learning [Sendera et al., 2023], and reinforcement learning [Sarafian et al., 2021].

### 3.3 H-score

H-score is firstly introduced by Huang et al. in 2019 as a metric assessing the informativeness of features for a task. Theoretically derived from the maximal correlation interpretation of deep neural networks, its mathematical foundation roots to the information theory work known as maximal correlation analysis, which originates from the works of Hirschfeld, Gebelein and Renyi [Hirschfeld, 1935, Gebelein, 1941, Rényi, 1959] and has been followed and further explored by a broad spectrum of successive work. The H-score of $f$ with regard to the task casting $X$ to $Y$ is defined as:

$$H(f) = tr(cov(f(X))^{-1} cov(\mathbb{E}_{P_{X|Y}}[f(X)|Y])), \tag{1}$$

with input data $X$, label $Y$ and feature extractor $f(X)$. Subsequent research has extended the application of H-score, establishing it as a metric for transferability and validating its efficiency through extensive experimental evaluations [Bao et al., 2019, Ibrahim et al., 2022]. This underscores H-score's potential for transfer learning and its relevance to related challenges. We have chosen to incorporate H-score into our framework due to its robust theoretical reliability, alignment of assumptions with our problem setting, and its independence from source data, enabling effective online embedding estimation.

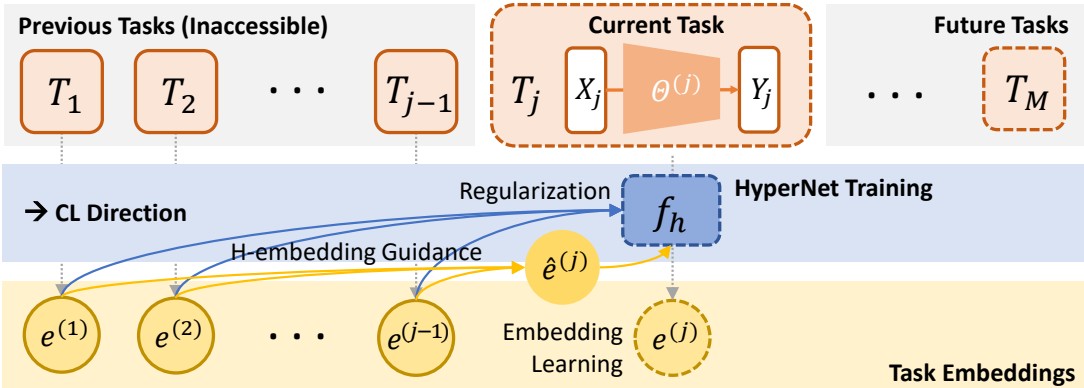

Figure 1: **Illustration of the CL status on the step of learning task $j$ under our framework.** The hypernet is being trained to provide the optimal task model weight $\Theta^{(j)}$ concurrently with the learning of current task embedding $e^{(j)}$, where regularization and guidance are applied using previous embeddings and H-embeddings.

## 4 Methodology

### 4.1 H-embedding

Unlike most existing approaches that focus on model elements, at the core of this work we incorporate task relationship information derived from accessible data into a prior embedding, and utilize it to guide a hypernet-based CL framework. To this end, we introduce H-embedding, a transferability-aware online task embedding built upon the H-score metric.[4] The acquisition of H-embedding is specially tailored to fit with the training process and data accessibility constraints in CL settings.

Specifically, during the training stage of task $j$, we first measure the H-score transferability from each previous task $\{T_n\}_{n=1}^{j-1}$ to $T_j$ using $D_j$ and previous task model parameters $\{\Theta^{(n)}\}_{n=1}^{j-1}$ by

$$H(T_n, T_j) = tr(cov(f_l(x^{(j)}, \Theta^{(n)}))^{-1} \cdot cov(\mathbb{E}_{P_{X|Y}}[f_l(x^{(j)}, \Theta^{(n)})|y^{(j)}])). \tag{2}$$

Here, $f_l(*)$ denotes the output of the last hidden layer in the task model $f$, which can be viewed as the feature of task data $X^{(j)}$.[5] We leverage the convenience that previous task models $\{\Theta^{(n)}\}_{n=1}^{j-1}$

---

[4]Our design of H-embedding and the adoption of H-score are primarily motivated by its computational efficiency, effectiveness in measuring transferability, and compatibility with the online CL setting; a more detailed discussion of task embedding alternatives is provided in Appendix A.3.

[5]*Note.* The first feature covariance is computed across samples $x$, and the second across class-conditional means over $y$.

can be reconstructed by the hypernet $f_h$ and corresponding task embeddings $\{e^{(n)}\}_{n=1}^{j-1}$, i.e. $\Theta^{(n)} = f_h(e^{(n)}, \Theta_h)$. The H-embedding $\hat{e}^{(j)}$ of current task is then computed by minimizing the difference between the Euclidean distance of $e^{(n)}, \hat{e}^{(j)}$ and their reversed H-score transferability $H(T_n, T_j)$:

$$\hat{e}^{(j)} = \arg\min_{\hat{e}^{(j)}} \sum_{n=1}^{j-1} \left( ||\hat{e}^{(j)} - e^{(n)}||_2 - 1/H(T_n, T_j) \right)^2, \tag{3}$$

where $\{e^{(n)}\}_{n=1}^{j-1}$ is calculated and stored when learning previous tasks. Given that assessing transferability requires a minimum of two tasks, H-embeddings and the guidance loss are computed only after completing the first two tasks.

Nevertheless, due to its target-centered intrinsicality, the absolute value of H-score is largely dependent on the characteristics of target task, resulting in asymmetry and inconsistent scaling across task pairs. This discrepancy prevents a direct alignment between the reversed H-score and the Euclidean distance among sequentially learned task embeddings. To alleviate this mismatch, we introduce Analytic Hierarchy Process (AHP) normalization [Zamir et al., 2018] to process the H-score in Eqn. 3. Specifically, we construct a pairwise tournament matrix $W^{(j)} \in \mathbb{R}^{j \times j}$ for task $j$, with elements given by:

$$w_{m,n}^{(j)} = \frac{H(T_m, T_j)}{H(T_n, T_j)} \quad \forall m, n \in \{1, 2, \ldots, j\}, \tag{4}$$

measuring how many times better task $m$ is compared to task $n$ when transferring to task $j$. Here, we define the self-transferability $H(T_j, T_j)$ as the HGR maximal correlation between $X_j$ and $Y_j$, with the consistency of transferability definition verified by the theoretical framework of H-score[6]. Consequently, the AHP normalized transferabilites are given by elements of the principal eigenvector $\mathbf{v}^{(j)}$ of $W^{(j)}$, i.e. $\mathcal{AHP}(T_n, T_j) = \mathbf{v}_n^{(j)}$, which could be reasonably converted to a distance metric through a negatively correlated affinity–distance transformation, $dist(T_n, T_j) = \gamma^{(j)} \exp(-\mathcal{AHP}(T_n, T_j))$. The scaling constant $\gamma^{(j)}$ would be optimized together with $\hat{e}^{(j)}$, modifying Eqn. 3 to:

$$\hat{e}^{(j)}, \gamma^{(j)} = \arg\min_{\hat{e}^{(j)}, \gamma^{(j)}} \sum_{n=1}^{j-1} (||\hat{e}^{(j)} - e^{(n)}||_2 - \gamma^{(j)} \exp(-\mathcal{AHP}(T_n, T_j)))^2. \tag{5}$$

Given that $H(T_n, T_j)$ and $e^{(n)}$ are directly calculated, the above optimization problem is a benign bi-variate optimization problem. We could thus apply a gradient descent algorithm to effectively compute the H-embedding $\hat{e}^{(j)}$ for the $j$-th task. As such, the H-embeddings for all tasks during the continual learning can be calculated in an inductive way.

### 4.2 H-Embedding Guided Hypernet

#### 4.2.1 Hypernet-Based CL Framework

Based on the task-relation-aware H-embedding, we leverage the critical role of task embeddings in hypernet-based models, and design a hypernet framework that incorporates H-embedding guidance into the task-specific parameter generation. In particular, we present a CL framework (illustrated in Fig. 1) where a task-conditioned hypernetwork $f_h(e, \Theta_h)$ with hypernet weights $\Theta_h$ is introduced to map task embeddings $\{e^{(j)}\}_{j=1}^M$ to the corresponding model weights $\{\Theta^{(j)}\}_{j=1}^M$ of task model $f$ for all CL tasks $\{T_j\}_{j=1}^M$, i.e., $\Theta^{(j)} = f_h(e^{(j)}, \Theta_h)$ for task $j$. When training on each task $T_j$, the task embedding $e^{(j)}$ is updated together with the optimization of hypernet parameters $\Theta_h$, while parameters other than $\Theta_h$ and $e^{(j)}$ are fixed and can be viewed as constants. The learning of $e^{(j)}$ and $\Theta_h$ is regularized using previous task embeddings $\{e^{(n)}\}_{n=1}^{j-1}$ and guided using H-embedding $\hat{e}^{(j)}$ to ensure backward and forward transfer performance. The learning loss is composed of three parts:

1) Target loss, a supervised loss to learn current task $j$

$$L_t = \mathcal{L}\left( f(x^{(j)}, \Theta^{(j)}), y^{(j)} \right) = \mathcal{L}\left( f(x^{(j)}, f_h(e^{(j)}, \Theta_h)), y^{(j)} \right). \tag{6}$$

---

[6]See Appendix A.2 for a brief proof and computation details.

2) Continual learning loss (following von Oswald et al. [2020]), to prevent CF by ensuring that given previous task embeddings $\{e^{(n)}\}_{n=1}^{j-1}$, the network weights output by the hypernet before and after the training on task $j$ are analogous

$$L_c = \frac{1}{j-1} \sum_{n=1}^{j-1} L_c^{(n)} = \frac{1}{j-1} \sum_{n=1}^{j-1} ||f_h(e^{(n)}, \Theta_h) - f_h(e^{(n)}, \Theta_h^*)||^2. \tag{7}$$

3) H-embedding guidance loss, to provide the hypernet with additional prior knowledge about the task relationships using transferability

$$L_e = L_e(e^{(j)}, \hat{e}^{(j)}). \tag{8}$$

Here, $\mathcal{L}$ denotes certain supervised task loss (cross-entropy loss in our experiments), and $\Theta_h^*$ is the set of hypernet parameters before learning task $j$. The definition of the embedding regularization loss $L_e$ will be covered in later sections. To summarize, our final loss function is as follows with hyperparameters $\beta_e$ and $\beta_c$:

$$L = L_t + \beta_e L_e + \beta_c L_c. \tag{9}$$

On the $j$-th task, our approach for the training of $T_j$ is depicted in Fig. 2. Notably, although it may appear that the task model weights are first generated and subsequently used for inference, the framework is actually end-to-end, with the hypernet parameters $\Theta_h$ and embeddings $e^{(j)}$ optimized directly by feeding the task data and minimizing the overall loss. Hence, there is no additional training procedure introduced in our framework, and the only information to save is the low-dimensional[7] task embedding $e^{(j)}$.

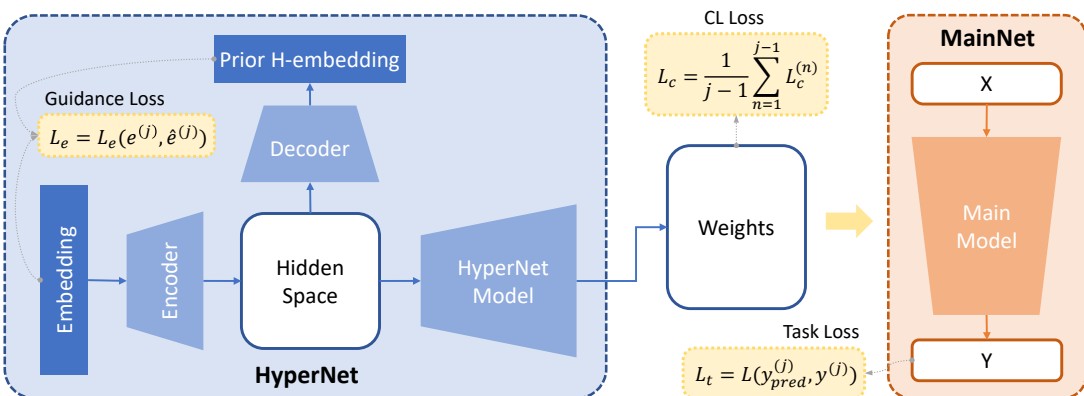

Figure 2: **Framework of our hypernet on the slice of task $j$.** A hypernet (left, blue) is utilized to learn the weights of the main model (right, orange), where the H-embedding guidance is introduced using an encoder-decoder module. The entire framework is trained end-to-end by inputting task data into the main model and propagating gradients backward to update both hypernet and embedding.

### 4.2.2  Embedding Guidance via Encoder and Decoder

In our framework, an embedding regularization module is introduced to integrate the H-embedding guidance into the hypernet. Specifically, we view the output of the hypernet's intermediate layer as a hidden representation $h$, and the preceding layers as an encoder $f_{Enc}$. From an information transmission perspective, it can be presumed that $h$ should retain sufficient information to recover the H-embedding $\hat{e}$. Therefore, we attach to the layer a lightweight trainable decoder $f_{Dec}$ that reconstructs an embedding $\tilde{e}$ from $h$, such that the discrepancy between $\tilde{e}$ and the H-embedding $\hat{e}$ should be minimized. Formally, for task $j$,

$$\tilde{e}^{(j)} = f_{Dec}(h^{(j)}) = f_{Dec}\left(f_{Enc}(e^{(j)})\right) \tag{10}$$

---

[7]The dimension of task embedding is set to 32 in our experiments.

| Backbone | Method | CIFAR-100 (N = 10) | | ImageNet-R (N = 10) | |
|---|---|---|---|---|---|
| | | FAA($\uparrow$) | DAA($\uparrow$) | FAA($\uparrow$) | DAA($\uparrow$) |
| ResNet-32 | Full Finetune | $19.13_{(1.35)}$ | $79.02_{(0.96)}$ | $15.08_{(0.62)}$ | $56.42_{(0.45)}$ |
| | LwF | $34.35_{(1.06)}$ | $84.55_{(0.27)}$ | $17.30_{(0.05)}$ | $40.66_{(0.27)}$ |
| | EWC | $37.83_{(2.58)}$ | $84.21_{(0.04)}$ | $15.77_{(0.29)}$ | $27.97_{(0.34)}$ |
| | L2 | $41.49_{(0.74)}$ | $84.96_{(0.18)}$ | $15.95_{(0.39)}$ | $63.77_{(0.56)}$ |
| | PredKD + FeatKD | $34.43_{(0.95)}$ | $83.72_{(0.90)}$ | $18.22_{(0.70)}$ | $40.37_{(0.75)}$ |
| | PackNet | $70.68_{(0.28)}$ | $70.68_{(0.28)}$ | $34.63_{(0.85)}$ | $34.63_{(0.85)}$ |
| | HyperNet | $81.57_{(0.41)}$ | $81.63_{(0.42)}$ | $38.03_{(1.21)}$ | $38.18_{(0.04)}$ |
| | WSN | $82.75_{(0.44)}$ | $82.75_{(0.44)}$ | $37.99_{(0.27)}$ | $37.99_{(0.27)}$ |
| | H-embed Hnet* | $\mathbf{83.08}_{(0.12)}$ | $83.09_{(0.09)}$ | $\mathbf{38.16}_{(1.13)}$ | $38.09_{(0.09)}$ |
| ViT-B/16 | Full Finetune | $69.49_{(0.50)}$ | $80.35_{(0.87)}$ | $60.57_{(1.06)}$ | $72.31_{(1.09)}$ |
| | L2P | $83.18_{(1.20)}$ | $87.69_{(1.05)}$ | $71.26_{(0.44)}$ | $76.13_{(0.46)}$ |
| | DualPrompt | $81.48_{(0.86)}$ | $86.41_{(0.66)}$ | $68.22_{(0.20)}$ | $73.81_{(0.39)}$ |
| | CODA-Prompt | $86.31_{(0.12)}$ | $90.67_{(0.22)}$ | $74.05_{(0.41)}$ | $78.14_{(0.39)}$ |
| | HiDe-Prompt | $93.48_{(0.11)}$ | $95.02_{(0.01)}$ | $74.65_{(0.14)}$ | $78.46_{(0.18)}$ |
| | InfLoRA | $86.75_{(0.35)}$ | $91.72_{(0.15)}$ | $74.75_{(0.64)}$ | $80.67_{(0.55)}$ |
| | SD-LoRA | $87.26_{(0.22)}$ | $92.05_{(0.31)}$ | $77.18_{(0.39)}$ | $81.74_{(0.24)}$ |
| | H-embed Hnet-LoRA* | $\mathbf{97.07}_{(1.61)}$ | $97.06_{(1.57)}$ | $\mathbf{81.38}_{(0.65)}$ | $81.67_{(0.10)}$ |

Table 1: **Accuracy (%) Comparison on CIFAR-100 (N = 10) and ImageNet-R (N = 10) on ResNet-32 and ViT-B/16 Backbones.** Our method (marked by '*') achieves the top FAA.

should approximate $\hat{e}^{(j)}$ as close as possible. Summing it up in a mathematical form, we have the embedding guidance loss for task $j$:

$$L_e = L_e(e^{(j)}, \hat{e}^{(j)}) = \mathcal{L}\left(f_{Dec}(f_{Enc}(e^{(j)})), \hat{e}^{(j)}\right). \tag{11}$$

$\mathcal{L}$ denotes a certain similarity criterion, set to the cosine similarity loss in our experiments. The H-embedding $\hat{e}^{(j)}$ is derived by Eqn. 5 and the decoder $f_{Dec}$ is updated together with the hypernet during training. Notably, no significant computing cost is posed with the introduction of the embedding regularization module given the encoder and decoder are both shallow fully connected neural networks. We summarize the training process of task $j$ as the algorithm in Appendix A.4.

### 4.2.3 Plug-in Application in PEFT Settings

While our previous formulations are mostly based on assuming that the hypernet generates the full parameter set $\Theta$ for the task model, this is not a strict requirement. In practice, our framework can be easily adapted to generate only a subset of the model parameters, making it naturally compatible with modern parameter-efficient fine-tuning (PEFT) techniques — such as prompt tuning and LoRA — which leverage frozen pretrained backbones and update only small, task-specific modules. Under this setting, our method can be readily deployed as a plug-in component that exclusively generates LoRA parameters, with the rest of the model kept fixed. We validate this lightweight variant in our experiments and demonstrate that it obtains strong performance, highlighting the flexibility, scalability, and practical utility of our approach.

## 5 Experiments

### 5.1 Experimental Settings

**Benchmarks.** To comprehensively verify the effectiveness of our framework and further analyze its reliability, we select three representative benchmarks from previous work on CL and perform extensive experiments on them: **PermutedMNIST** (10 tasks) [Goodfellow et al., 2013], **CIFAR-100** (10 tasks) [Krizhevsky et al., 2009], **DomainNet** (5 tasks) [Peng et al., 2019] and **ImageNet-R** (5

tasks, 10 tasks and 20 tasks) [Hendrycks et al., 2021][8]. A detailed description of these benchmarks is listed in Appendix A.5.

**Evaluation Metrics.** Following previous works [Qu et al., 2021, Wang et al., 2024], we evaluate the different CL methods with two widely adopted metrics. Final Average Accuracy: $FAA = \frac{1}{M}\sum_{j=1}^{M} a_{j,M}$, the average task accuracy measured after the completion of learning all $M$ tasks; During Average Accuracy: $DAA = \frac{1}{M}\sum_{j=1}^{M} a_{j,j}$, the average accuracy of each task measured immediately after it is learned. Here, $a_{i,j}$ denotes the accuracy (%) measured on the test set of $i$-th task after learning the $j$-th task. In terms of CL analysis, FAA provides a summary evaluation of overall CL performance, while DAA offers deeper insights into the forward/backward transfer behavior and model learning dynamics.

## 5.2 Performance Evaluation

**Comparison Experiments with Baselines from Multiple Paradigms.** Our primary evaluation study is conducted on the CIFAR-100 and ImageNet-R benchmarks, both consists of 10 tasks. To ensure fairness and comprehensiveness in comparison, a ResNet-32 [He et al., 2016] and a pretrained ViT-B/16 [Dosovitskiy et al., 2020] are selected as the shared backbone models for the full-model and parameter-efficient methods respectively. All reported results are either obtained by our own implementation (averaged over three random seeds with standard deviation) or directly cited from previous works using the same benchmark and backbone.

The choice of baselines is based on the requirements that they should both conform to a rehearsal-free setting and be applicable to these benchmarks and backbones. For a thorough comparison with existing methods to the greatest extent possible, we select representative baselines of varied methodology categories. Full-model strategies include: **Regularization Methods:** LwF [Li and Hoiem, 2017], EWC [Kirkpatrick et al., 2017], L2, PredKD+FeatKD [Smith et al., 2023b]; **Architecture Methods:** PackNet [Mallya and Lazebnik, 2018], HyperNet [von Oswald et al., 2020], WSN [Kang et al., 2022]. Parameter-efficient finetuning (PEFT) based strategies include: **Prompt Methods:** L2P [Wang et al., 2022b], DualPrompt [Wang et al., 2022a], CODA-Prompt [Smith et al., 2023a], HiDe-Prompt [Wang et al., 2023]; **LoRA Methods:** InfLoRA [Liang and Li, 2024], SD-LoRA [Wu et al., 2025]. Full finetuning is also included as a basic reference in both backbones. For our framework, we construct full model and LoRA generation versions on ResNet and ViT respectively. The experimental results are summarized in Tab. 1, where our method achieves the highest FAA across both backbones and benchmarks, demonstrating superior overall CL performance. In addition, it attains a competitive DAA that closely aligns with FAA, suggesting strong forward and backward transfer capabilities.

**Extended Evaluation of PEFT Methods under Varying Task Numbers and Challenging Benchmarks.** Given the stronger model capacity and CL performance typically exhibited by PEFT-based methods through leveraging pretrained models, we conduct additional evaluations to further assess their robustness under more challenging settings, including varying task sequence lengths and the more difficult DomainNet benchmark. As shown in Tab. 2, our method consistently outperforms all baselines. Notably, its advantage becomes more pronounced as the number of tasks increases, highlighting its robustness in scenarios with longer and more challenging task sequences.

**Ablation Studies.** To take a better concentration on validating our design of framework, we conduct ablation studies on ImageNet-R (N = 5, 10, 20) benchmarks with ViT-LoRA backbone, and summarize the results in Fig. 3. The comparison baselines are: without H-embedding guidance (w/o Hemb), without CL regularization (w/o CLreg), without AHP normalization (w/o AHP), and SD-LoRA, the SOTA baseline for reference. For a more comprehensive evaluation, we also conduct extra ablation studies on full model generation under different backbones and benchmarks, covering PermutedMNIST, CIFAR-100 and ImageNet-R, with results list in Appendix A.7. A notable increase in CL performance could be observed across all benchmarks and backbones.

---

[8]Training specifics of our experimental studies are listed in Appendix A.6, with codes available at `https://github.com/viki760/Hembedding_Guided_Hypernet`.

| Method | ImageNet-R (N = 5) | | ImageNet-R (N = 20) | | DomainNet (N = 5) | |
|--------|--------|--------|--------|--------|--------|--------|
| | FAA↑ | DAA↑ | FAA↑ | DAA↑ | FAA↑ | DAA↑ |
| Full Finetune | $64.92_{(0.87)}$ | $75.57_{(0.50)}$ | $49.95_{(1.31)}$ | $65.32_{(0.84)}$ | $51.46_{(0.47)}$ | $67.08_{(1.13)}$ |
| L2P | $73.04_{(0.71)}$ | $76.94_{(0.41)}$ | $68.97_{(0.51)}$ | $74.16_{(0.32)}$ | $70.26_{(0.25)}$ | $75.83_{(0.98)}$ |
| DualPrompt | $69.99_{(0.57)}$ | $72.24_{(0.41)}$ | $65.23_{(0.45)}$ | $71.30_{(0.16)}$ | $68.26_{(0.90)}$ | $73.84_{(0.45)}$ |
| CODA-Prompt | $76.63_{(0.27)}$ | $80.30_{(0.28)}$ | $69.38_{(0.33)}$ | $73.95_{(0.63)}$ | $70.58_{(0.53)}$ | $76.68_{(0.44)}$ |
| HiDe-Prompt | $74.77_{(0.25)}$ | $78.15_{(0.24)}$ | $73.59_{(0.19)}$ | $77.93_{(0.19)}$ | $72.20_{(0.08)}$ | $77.01_{(0.04)}$ |
| InfLoRA | $76.95_{(0.23)}$ | $81.81_{(0.14)}$ | $69.89_{(0.56)}$ | $76.68_{(0.57)}$ | $71.59_{(0.23)}$ | $78.29_{(0.50)}$ |
| SD-LoRA | $79.01_{(0.26)}$ | $82.50_{(0.38)}$ | $74.05_{(0.51)}$ | $80.65_{(0.35)}$ | $72.58_{(0.40)}$ | $78.79_{(0.78)}$ |
| H-embed Hnet-LoRA* | $\mathbf{79.27}_{(0.42)}$ | $79.72_{(0.28)}$ | $\mathbf{79.90}_{(2.54)}$ | $84.79_{(0.37)}$ | $\mathbf{76.64}_{(0.95)}$ | $78.26_{(0.37)}$ |

Table 2: **Further Evaluation on ImageNet-R (N = 5, 20) and DomainNet (N = 5).** Our method (marked by '*') consistently obtains the highest FAA.

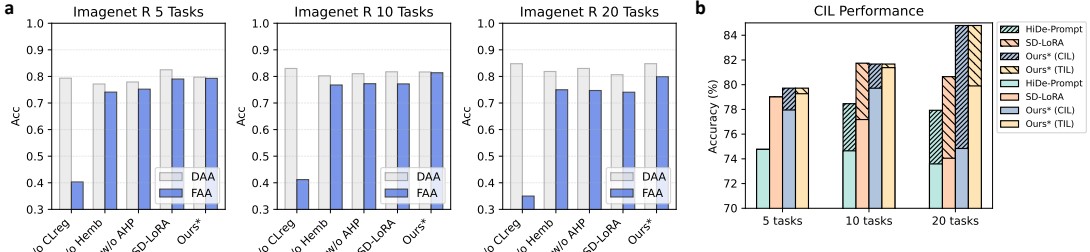

Figure 3: **Illustration of Ablation Studies (a) and CIL Performance (b).** Left (a): FAA and DAA results of ablation studies (averaged across seeds). Right (b): FAA and DAA (striped) results of CIL baselines.

## 5.3 Discussion and Further Performance Analysis

**Optimal Overall Transfer Ability.** As shown in previous results, our method demonstrates strong overall performance across various CL benchmarks. To gain deeper insight into its effectiveness, we analyze the framework from the perspective of forward transfer (FWT) and backward transfer (BWT) — two essential indicators in CL. Formally, forward transfer is defined as FWT := DAA - GTA, and backward transfer as BWT := FAA - DAA, where GTA refers to the ground-truth accuracy obtained by individually training a backbone model on each task. Higher values on both metrics indicate better CL capability. As illustrated in Fig. 3, which provides a visually intuitive comparison of the discrepancies between FAA and DAA across baselines, our method demonstrates competitive abilities in both FWT and BWT, thereby displaying prominent CL performance. Due to the space limit, we include detailed numerical results on the metrics in Appendix A.8.

**Parameter Size and Efficiency.** In our framework design, the overall number of hypernet parameters (including task embeddings) is strictly constrained to be no larger than that of the main network. Both the decoder module and H-embedding are lightweight, implemented as a two-layer MLP and a 32-dimensional vector respectively. This compact design ensures that the H-embedding guided hypernet does not introduce parameter overhead even with long task sequences. The storage efficiency is also preserved, as only the hypernet and task embedding need to be stored.

In terms of time efficiency, a single prediction in the hypernet framework involves one forward pass through the hypernet and one through the main net, theoretically doubling the inference time consumption. However, in practice the hypernet inference is executed only once per task to generate task-specific weights, and its time cost is amortized across all samples. As a result, the per-sample overhead becomes negligible on large datasets. The ViT-LoRA setting is even more efficient, where the hypernet only generates compact LoRA parameters rather than full model weights. We conduct time tests for both the ResNet backbone and the ViT-LoRA backbone, and the results support our analysis: on CIFAR100, vanilla ResNet32 takes 4.257s to process the full test set, compared to 4.260s with our framework; on ImageNet-R, vanilla ViT takes 4.313s, compared to 4.568s with our ViT-LoRA framework. Here a bit more time cost is introduced by the addition of LoRA modules.

**TIL versus CIL.** Up to now, our main experiments are conducted under the task-incremental learning (TIL) setting, where task identifiers are provided during both training and inference. However, in class-incremental learning (CIL), the unavailable task ID at test time can make the problem more challenging. Hopefully, due to the orthogonality of our framework with task ID inference, we could adapt it to the CIL setting by introducing an auxiliary module to infer the task ID. Specifically, during training, we cache the intermediate features of input samples extracted from a frozen pretrained model to train a task ID classifier, and use it to infer the task IDs during testing, and select the corresponding task embedding.We evaluate this extension on ImageNet-R (N = 5, 10, 20). As shown in Fig. 3, our CIL method remains competitive with strong baselines, demonstrating its applicability beyond the TIL setting. Nevertheless, this current approach is admittedly limited by its reliance on pretrained model features and the separately trained task classifier, which does not align tightly with the online, continual nature of our framework. We will continue to explore more integrated and dynamic task ID inference strategies that can be better coupled with our architecture in the future.

## 6    Conclusion

In this work, we propose a transferability-aware task embedding guided hypernet to exploit the task relationships for continual learning. By introducing the information theoretical transferability based task embedding named H-embedding and incorporating it in a hypernetwork, we establish an online framework capable of capturing the statistical relations among the CL tasks and leveraging this knowledge for guiding task-conditioned model weight generation. Through extensive experimental studies, we validate that the adoption of H-embedding guidance enhances continual learning by facilitating inter-task transfer and improving the reliability of task embeddings, achieving the best final accuracy performance under various CL benchmarks.

## Acknowledgments and Disclosure of Funding

This work is supported in part by the Natural Science Foundation of China (Grant 62371270) and Shenzhen Key Laboratory of Ubiquitous Data Enabling (No.ZDSYS20220527171406015).

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

# A Technical Appendices and Supplementary Material

## A.1 Continual Learning Setting

### A.1.1 Rehearsal-Free CL

The strictness of CL settings varies with the extent of allowed previous data accessibility. Multi-task learning [Caruana, 1997], with full data availability of all tasks, can actually be viewed as a special case of CL, while rehearsal-free CL [Smith et al., 2023b], with no previous data involved in the training of new tasks, is the strictest CL setting under this criteria. Despite the success of rehearsal-based methods in various benchmarks [Bang et al., 2021, Shin et al., 2017, Belouadah and Popescu, 2019], rehearsal-free CL is catching the attention of researchers recently [Smith et al., 2023b] because of its low dependency on revisiting previous tasks and therefore broader application in the era of growing data privacy concerns. Existing works on rehearsal-free CL are mostly based on regularization strategies. EWC [Kirkpatrick et al., 2017] and SI [Zenke et al., 2017] introduce penalties to restrict the alteration of parameters vital for addressing prior tasks, thereby reducing the risk of CF. LwF [Li and Hoiem, 2017, Rebuffi et al., 2017] proposes a cross-entropy loss between the predicted class distribution of the *(n-1)*-th task, as generated by the model before and after learning the *n*-th task. Smith et al. [2023b] reviews these methods and proposes regularization combinations for better CL performance. In this work, we follow these works and focus on the more challenging rehearsal-free CL setting.

### A.1.2 TIL versus CIL

Based on the discrepancy between $D_{j-1}$ and $D_j$, Hsu et al. [2018] and Van de Ven and Tolias [2019] categorize CL settings into three specific scenarios: task incremental, class incremental, and domain incremental. Table 3 summarizes the differences among these scenarios. For a better concentration on the study of CL methodology, our work mainly focuses on the task incremental CL (TIL) . In this scenario, the output spaces of tasks are partitioned by task IDs and mutually exclusive between $D_{j-1}$ and $D_j$, which is denoted as $\mathbf{Y}^{(j-1)} \neq \mathbf{Y}^{(j)}$. It can be then naturally indicated that $P(\mathbf{Y}^{(j-1)}) \neq P(\mathbf{Y}^{(j)})$ and $P(\mathbf{X}^{(j-1)}) \neq P(\mathbf{X}^{(j)})$. Notably, here task IDs are accessible during both training and testing. An adaptation of our method to other CL settings could be effectively conducted by introducing additional task inference modules similar to previous works. As an instance, we made a trial in this work that adapts our framework to class incremental CL (CIL) in Sec. 5.3 using a task ID classifier trained on image features.

| Scenario | $P(\mathbf{X}^{(j-1)}) \neq P(\mathbf{X}^{(j)})$ | $P(\mathbf{Y}^{(j-1)}) \neq P(\mathbf{Y}^{(j)})$ | $\mathbf{Y}^{(j-1)} \neq \mathbf{Y}^{(j)}$ | Task ID |
|---|---|---|---|---|
| Domain Incremental | ✓ | ✗ | ✗ | ✗ |
| Class Incremental | ✓ | ✓ | ✗ | ✗ |
| Task Incremental* | ✓ | ✓ | ✓ | ✓ |

Table 3: **Categorization of CL settings based on the discrepancy between $D_{j-1}$ and $D_j$.** '*' denotes the scenario focused on in our work.

## A.2 HGR maximal correlation for self transferability $H(T_j, T_j)$

As the theoretical basis of H-score transferability, the Hirschfeld–Gebelein–Rényi (HGR) maximal correlation of random variables $X$ and $Y$ over alphabets $\mathcal{X}$ and $\mathcal{Y}$ is defined as:

$$\mathcal{HGR}(X,Y) = \max_{\substack{f:\mathcal{X}\to\mathbb{R}^k, \, g:\mathcal{Y}\to\mathbb{R}^k \\ \mathbb{E}[f(X)]=\mathbb{E}[g(Y)]=\mathbf{0} \\ \mathbb{E}[f(X)f^\top(X)]=\mathbb{E}[g(Y)g^\top(Y)]=\mathbf{I}}} \mathbb{E}_{P_{XY}} \left[ f^\top(X)g(Y) \right]. \tag{12}$$

Here, the correlation is derived by taking the maximum over all functions $f$, $g$ with zero mean and unit variance, which hence extracts the most correlated aspects of $X$ and $Y$. This definition is equivalent to the maximum of the two-sided H-score given by Huang et al. [2019] with normalized functions $f$ and $g$:

$$H(f,g) = \mathbb{E}_{P_{XY}} \left[ f^\top(X)g(Y) \right] - \frac{1}{2}tr(cov(f(X))cov(g(Y))), \tag{13}$$

with the one-sided H-score extended from Eqn. 13 by assuming the function $g$ as optimal:

$$H(f) = tr(cov(f(X))^{-1}cov(\mathbb{E}_{P_{X|Y}}[f(X)|Y])). \tag{14}$$

Notably, Eqn. 14 is exactly the definition of H-score transferability [Bao et al., 2019], which is actually proposed by applying the H-score theories to transfer learning scenarios. Therefore, with normalized $f$, $g$, the HGR maximal correlation between $X_j$, $Y_j$ is mathematically equivalent to the H-score metric of the theoretical optimal model $f_j^*$ on task $j$.

In our work, we utilize the H-score transferability metrics denoted as $H(T_i, T_j)$ for tasks $i \neq j$. However, the H-score for task $j$ cannot be computed while the model remains untrained for that specific task. To maintain consistency in our definitions, we define $H(T_j, T_j)$ as the HGR maximal correlation between the input $X_j$ and the output $Y_j$, implicitly leveraging the theoretical optimal model $f_j^*$ for task $j$. Specifically, in our computations, the models $f$ and $g$ are implemented as fully connected neural networks equipped with normalization layers, and they undergo 100 epochs of training using a sampled subset of the training data through gradient descent.

### A.3   Discussion on the Use of H-Score and Alternative Task Embeddings

While technically task embeddings can be estimated using various methods, the alternatives are generally less suitable for our target setting, as they either require access to source data for each task—assumptions that are infeasible under rehearsal-free CL—or do not explicitly model task-to-task relationships, thereby offering limited benefits in facilitating bidirectional transfer across CL tasks.

To address this, we propose a new formulation where task embeddings are computed online by explicitly optimizing them to approximate a transferability metric across tasks. In this design, we adopt H-score as the target transferability metric primarily due to its source-data-free nature, which allows us to maintain compliance with CL constraints. Additionally, H-score is theoretically grounded, computationally efficient, and empirically shown to correlate well with actual transfer performance. These characteristics make it a particularly suitable and representative choice for our embedding algorithm. Yet, our framework is not inherently tied to H-score, and there admittedly exist other source-free embeddings or transferability metrics (e.g. Task2Vec [Achille et al., 2019], LEEP [Nguyen et al., 2020], LogME [You et al., 2021]) that could also, in principle, be used within it.

As both an ablation and a further assessment of our framework's flexibility, we conducted preliminary experiments on ImageNet-R 10 task benchmark by:

- Replacing H-score with other source-free transferability metrics (LEEP and LogME) while keeping the embedding algorithm and overall pipeline unchanged.
- Replacing the entire H-embedding formulation with Task2Vec.

The results are shown in Tab. 4.

| Method | H-Embedding* | H-Embedding (LEEP) | H-Embedding (LogME) | Task2Vec |
|---|---|---|---|---|
| FAA (%) | 81.8 | 81.6 | 81.4 | 79.8 |

Table 4: **Comparison of different task embedding variants on the ImageNet-R 10-task benchmark.**

As shown, replacing the transferability metric alone leads to only minor drop in performance. This supports our view that while H-score is a strong instantiation, the core contribution of our work lies in the transferability-guided embedding formulation and its integration into a hypernetwork-based parameter generation pipeline—a general framework that remains valid even when alternative metrics are employed.

However, replacing the entire embedding formulation with task2vec leads to a more noticeable performance degradation. We believe this is because Task2Vec computes task embeddings independently without explicitly modeling task-to-task relationships, which limits its capacity to guide transfer dynamics in CL. Additionally, Task2Vec is highly sensitive to the choice and training of the probe network, which introduces extra variance in effectiveness and incurs additional memory costs—challenges not present in our H-embedding approach.

## A.4 Algorithm for H-embedding Guided Hypernet Framework

For a better understanding of our framework, we summarize the entire training process of task $j$ within our H-embedding guided hypernet as outlined in Algorithm 1.

---

**Algorithm 1:** H-embedding guided Hypernet: Training of Task $j$

---

**Input:** Task data $D_j$, previous task embeddings $\{e^{(n)}\}_{n=1}^{j-1}$, hypernet weights $\Theta_h$
**Parameter:** Learning rate $\lambda$
**Output:** Current task embedding $e^{(j)}$, updated hypernet weights $\Theta_h$
Randomly initialize $e^{(j)}, \hat{e}^{(j)}$;
**if** $j > 2$ **then**
    **for** $n \leftarrow 1$ **to** $j-1$ **do**                              // Compute transferability
        $\Theta^{(n)} \leftarrow f_h(e^{(n)}, \Theta_h)$ ;
        $H(T_n, T_j) \leftarrow tr\left(cov(f_l(x^{(j)}, \Theta^{(n)}))^{-1} cov(\mathbb{E}_{P_{X|Y}}[f_l(x^{(j)}, \Theta^{(n)})|y^{(j)}])\right)$     $\triangleright$ Eq. 2
    **end**
    Randomly initialize $\gamma^{(j)}$;
    $\hat{e}^{(j)}, \gamma^{(j)} \leftarrow \arg\min_{\hat{e}^{(j)}, \gamma^{(j)}} \sum_{n=1}^{j-1}\left(||\hat{e}^{(j)} - e^{(n)}||_2 - \gamma^{(j)}\exp(-\mathcal{AHP}(T_n, T_j))\right)^2$ $\triangleright$ Eq. 5
**end**
**repeat**                                                  // Train hypernet
    $e^{(j)} \leftarrow e^{(j)} - \lambda\nabla_{e^{(j)}}L$;
    $\Theta_h \leftarrow \Theta_h - \lambda\nabla_{\Theta_h}L$                               $\triangleright$ Eq. 9
**until** *converge*;
**Return** $e^{(j)}, f_h(\,\cdot\,, \Theta_h)$

---

## A.5 Benchmarks of Experiments

**PermutedMNIST** [Goodfellow et al., 2013] (N=10) benchmark is a variant of MNIST [LeCun et al., 1998], forming CL tasks from the original MNIST dataset by applying random permutations to the input image pixels. The permuting procedure can be repeated 9 times in experiments to yield a task sequence of 10, with each task consisting of 70,000 images (60,000 for training and 10,000 for testing) of digits from 0 to 9. **CIFAR-100** [Krizhevsky et al., 2009] (N = 10) is a benchmark composed of 10 ten-way classification tasks composed by splitting a CIFAR-100 dataset into ten tasks. The model is sequentially trained on the tasks, each with 6,000 images (5,000 for training and 1,000 for testing). **ImageNet-R** [Hendrycks et al., 2021], built upon the ImageNet dataset [Deng et al., 2009], features a diverse range of renditions of ImageNet classes. This benchmark includes a total of 30,000 images across 200 classes from ImageNet. For continual learning evaluation, ImageNet-R (N = 5, 10, 20) is formed by organizing ImageNet-R into 5, 10, and 20 tasks respectively, each containing 40, 20, and 10 classes and around 6000, 3000, 1500 samples (roughly 5/6 for training and 1/6 for testing). **DomainNet** [Peng et al., 2019] (N = 5) is a benchmark derived from the 345 classes classification task across six distinct domains, by fusing all domains and splitting into 5 * 69 classification tasks.

## A.6 Experimental settings

### A.6.1 Comparison Experiments (Table 1)

**Choice of Baselines.** Our selection of baselines in this work aims to encompass a wide range of baseline categories, covering two of three primary categories in contemporary CL researches (*i.e.*, replay-based, regularization-based, and architecture-based), with the replay-based methods not conforming to our rehearsal-free setting. The specific choice of baselines in each category is mainly based on performance comparison conclusions in recent works such as Smith et al. [2023b] and Kang et al. [2022]. Therefore, we believe that our comparison study has included the most competitive and representative baselines.

**General Settings.** In CIFAR-100 and ImageNet-R(10 tasks) datasets with the ResNet-32 backbone network, we evaluate various baseline methods applicable to its model structure including Full Finetune, EWC, L2, PredKD+FeatKD, PackNet, HyperNet, WSN. All baselines are implemented

on our own using a ResNet-32 with 'option A', *i.e.*, leveraging the zero-padding shortcuts for increasing dimensions. In CIFAR-100 and ImageNet-R datasets with the ViT-B/16 backbone network, we mainly evaluate PEFT baseline methods including L2P, DualPrompt, CODA-Prompt, HiDe-Prompt, InfLoRA, and SD-LoRA. We cited the results of most baselines from previous works [Wang et al., 2023, Wu et al., 2025]. For ResNet, the experiments on CIFAR-100 are conducted on NVIDIA GeForce RTX 3090 GPUs with 100 epochs of training (unless early-stop), and the ImageNet-R experiments are carried out on NVIDIA A800 or A100 GPUs with 200 epochs of training (unless early-stop). For ViT, all experiments are conducted on NVIDIA A800 or A100 GPUs until convergence.

To ensure a fair comparison, we adopt consistent training settings across all baseline methods we implement (unless listed separately). Specifically, the batch size is set to 32, and we use the Adam optimizer with an initial learning rate of 0.001. The learning rate is decayed by a factor of 10 after the 50th and 75th epochs. A weight decay of $1 \times 10^{-4}$ is applied. For robustness, each experiment is run three times with different random seeds 22, 32, and 42, and the results are averaged.

**Details of Specific Methods.** For the **Finetune** baseline, the model is sequentially trained on each task without any mechanisms to prevent catastrophic forgetting. The model is randomly initialized and trained from scratch on the first task. The training of subsequent tasks continues using the weights obtained from the previous tasks.

For the **EWC** baseline [Kirkpatrick et al., 2017], we add a regularization term to the loss function to penalize significant changes to parameters important for previously learned tasks. The importance of each parameter is estimated using the Fisher Information Matrix. The regularization coefficient $\lambda$ is set to 10, following standard practice.

In the **L2** baseline, an L2 regularization term is added to the loss function to limit changes in the model parameters during training on new tasks. The regularization coefficient $\lambda$ is set to 1.0, determined by tuning on a small validation set derived from the training data of the first task.

For the **PredKD + FeatKD** method [Smith et al., 2023b], we incorporate both prediction distillation and feature distillation to transfer knowledge from previous tasks to new ones. The distillation loss combines the Kullback-Leibler divergence between the soft outputs of the teacher (model trained on previous tasks) and the student (current model), as well as the mean squared error between their intermediate feature representations. The loss weights are set to $\alpha = 1.0$ and $\beta = 0.5$ based on preliminary tuning.

In the **PackNet** method [Mallya and Lazebnik, 2018], we employ iterative pruning to allocate dedicated network weights for each task. After training on each task, we prune a certain percentage of the weights with the smallest magnitudes. Following the recommendations in the original paper, we experiment with pruning rates of 0.5, 0.75, and 0.8. We select the pruning rate of 0.8, which yields the best performance in our setting. After pruning, we fine-tune the remaining weights for an additional 10 epochs with a reduced learning rate of $1 \times 10^{-4}$.

For **HyperNet** method [von Oswald et al., 2020], we primarily follow the original work in training settings, with the learning rate being 0.001, the CL loss beta being 0.05, and scheduling and transforming strategies being the same as those used by Oswald. The embedding dimension is set to 32.

For the **WSN** method [Kang et al., 2022], we follow its original paper and use the default values of parameters in its official code repository. We choose the sparsity parameter $c = 0.5$ which performs best as listed in the WSN literature. Other parameters are set to the following values: optimization via Adam, a learning rate initialized at 1e-3 with a minimum of 1e-6, and a patience of 6 epochs for reducing the learning rate by a factor of 2. The models are trained for 100 epochs, with a batch size of 64 for both training and testing.

For **H-embedding guided hypernet**, the learning rate is set to 0.0005, with the embedding loss beta and CL loss beta both set to 0.05. The scheduling and transforming strategies are set the same as those of von Oswald et al. [2020] and the embedding dimension is also set to 32. For the learning of H-embedding, we update Eqn. 5 using gradient descent for 2000 iterations and the $f$, $g$ in HGR maximal correlation for 100 epochs respectively, both using a subset of 1000 samples from the training set.

For the **H-embedding guided hypernet with LoRA**, the learning rate is set to 0.001, with both the embedding loss beta and CL loss beta set to 0.05. The LoRA rank and alpha are configured at 16, resulting in about 0.7 million parameters generated by the hypernetwork, which is only 0.86% of the full ViT model's 86 million parameters. The LoRA dropout rate is set to 0.1. For CIFAR-100, a batch size of 32 and 15 epochs are used, while for ImageNet-R, a batch size of 128 and 60 epochs are used.

In all methods, we adhere to the principles of continual learning by not tuning hyperparameters on the full task set. In hypernet generation, special care was taken in handling batch normalization layers, especially in methods involving parameter freezing or pruning. We store and update batch normalization statistics separately for each task to ensure proper normalization during both training and inference. Also, to ensure that the parameter size of the hypernet framework should not exceed that of a normal network, we follow von Oswald et al. [2020] and chunk the weights into size of 7000, using additional chunking embedding to generate each chunked weight.

### A.6.2 Additional Comparison Experiments (Table 2)

**Hyperparameters with Change of Task Length.** In these experiments, we vary the number of tasks and adjusted the hyperparameters accordingly to evaluate their impact on performance. For scenarios involving five tasks, we set the training to span 60 epochs with a beta value of 0.006. Conversely, when the number of tasks is increased to 20, the training is conducted over 30 epochs with an augmented beta value of 0.05. These adjustments are designed to optimize learning by accommodating the varying complexities introduced by different task lengths.

**DomainNet Experiments.** In the DomainNet settings, the number of tasks also influences the hyperparameter configuration. For experiments involving five tasks, training is executed over 40 epochs with a beta of 0.0025 to ensure adequate learning across diverse domains. When expanded to ten tasks, the training epoch is reduced to 15, concurrently increasing beta to 0.05.

### A.7 Ablation Studies

### A.7.1 H-embed Hnet - LoRA (ViT) Ablation

Due to space limit in main text, we list the results of ablation studies on ViT-LoRA backbone here in Table 5. The experiments are conducted with the same hyperparameter setting as in comparison studies.

### A.7.2 H-embed Hnet - Full Generation Ablation

To broaden the comprehensiveness of evaluation and take a better concentration on validating our introduction of H-embedding guidance, we conduct extra ablation studies on three differed settings with different benchmarks as well as model backbones. Namely, experimental settings include: PermutedMNIST (10 tasks) using an MLP model, CIFAR-100 (10 tasks) using a 4-layer CNN model, and ImageNet-R (10 tasks) using a ResNet-32 model. We compared across three methods where all hyperparameters are set exactly to the same: 1) Vanilla Hnet, the hypernet CL framework without guidance module; 2) Rand-embed Hnet, the same framework as ours but replacing H-embedding with a random embedding; 3) H-embed Hnet, our framework. The performance is evaluated and summarized in Table 6, where a broad increase in CL performance could be observed across all benchmarks and backbones. Detailed information is listed below.

**PermutedMNIST.** Considering the smaller data dimension and model size in this setting, the embedding size is reduced to 24 and the training iteration number is set to 5000. The backbone model on PermutedMNIST is selected to be an MLP with fully-connected layers of size 1000, 1000 as used by Van de Ven and Tolias [2019]. We configure the learning rate as 0.0001 and the embedding loss beta as 0.05. The results are derived on NVIDIA GeForce RTX 3090 GPUs.

**CIFAR-100.** The tasks in this setting are derived the same as in comparison studies. Yet, the backbone model is differently set to a 4-layer CNN as used by Zenke et al. [2017]. We also follow Oswald in most of the hyperparameters, configuring the learning rate to 0.0001, embedding size to 32, as well as using the same scheduling strategies. We train each method with 100 epochs and the

embedding loss beta is set to 0.2 for H-embed and rand-embed hypernets. The results are derived on NVIDIA GeForce RTX 3090 GPUs.

**ImageNet-R.** For the ImageNet-R dataset, we split the original 200 classes into ten 20-way classification tasks. Because of the uneven class sample size of the ImageNet dataset, each task has varied numbers of training and test samples: Task 1 with 2,166 training samples and 543 test samples, Task 2 with 2,655 training and 716 test samples, . . . until Task 9 with 2,058 training samples and 471 test samples. In our method, we use a learning rate of 0.0005 and an embedding loss beta of 0.05, training the models for 200 epochs. The backbone model is the same as used in comparison experiments. The results are derived on NVIDIA A100 GPUs.

| Method | ImageNet-R (N = 5) | | ImageNet-R (N = 10) | | ImageNet-R (N = 20) | |
|---|---|---|---|---|---|---|
| | FAA↑ | DAA↑ | FAA↑ | DAA↑ | FAA↑ | DAA↑ |
| w/o CLreg | 39.08 | 79.93 | 39.89 | 83.21 | 34.57 | 85.11 |
| w/o Hemb | 77.37 | 78.56 | 42.85 | 82.05 | 74.82 | 83.55 |
| w/o AHP | 76.37 | 77.23 | 78.37 | 80.02 | 74.72 | 83.47 |
| Ours* | **79.72** | 80.10 | **81.65** | 81.86 | **76.91** | 85.05 |

Table 5: **Ablation Studies of H-embed Hnet - LoRA (ViT) on ImageNet-R (N = 5, 10, 20).** Our framework derives the best performance on all three task lengths.

| Setting | PermutedMNIST | | CIFAR-100 | | ImageNet-R | |
|---|---|---|---|---|---|---|
| | *MLP* | | *CNN* | | *ResNet-32* | |
| Method | FAA↑ | DAA↑ | FAA↑ | DAA↑ | FAA↑ | DAA↑ |
| Vanilla Hnet | 97.495 | 97.488 | 69.620 | 79.490 | 38.202 | 38.307 |
| Rand-embed Hnet | 97.448 | 97.447 | 71.250 | 76.490 | 38.046 | 37.956 |
| H-embed Hnet* | **97.553** | 97.560 | **72.390** | 76.900 | **39.212** | 39.299 |

Table 6: **Ablation Study of H-embed Hnet - Full Generation on Different Benchmarks and Backbones.** Our H-embedding guidance proves to be effective across all three settings, attaining the highest FAA and DAA.

### A.8  In-depth Performance Analysis

For a better analysis of the effectiveness of our strategy, we further investigate the detailed training behavior displayed in CL strategies, showing that our H-embedding guided hypernet is characterized by the following superiority.

**Quicker Convergence.** With the intention of understanding how our guidance aids the training process, we visualize the test accuracy trends during the training stage of tasks 3, 6, 10 of the 10 CL tasks under Cifar-ResNet setting in Fig. 4. It is shown in the figures that, compared to a hypernet without H-embedding guidance, our method converges noticeably faster and achieves a higher final accuracy performance, especially with the growth of task numbers. Such a phenomenon serves as a further suggestion that our H-embedding guidance provides a substantial enhancement to the task learning in CL through forward transfer.

**Embedding Interpretability.** To assess the task embeddings $\{e^{(j)}\}_{j=1}^{M}$ learned in our framework, we compute the task-wise Euclidean distances of the embeddings obtained with and without H-embedding guidance, and visualize the discrepancy between these two distance matrices in Fig. 5. Red signifies that the with-guidance embeddings result in a closer distance between the two tasks compared to the without-guidance embeddings, while blue represents the opposite. Take task 8, a

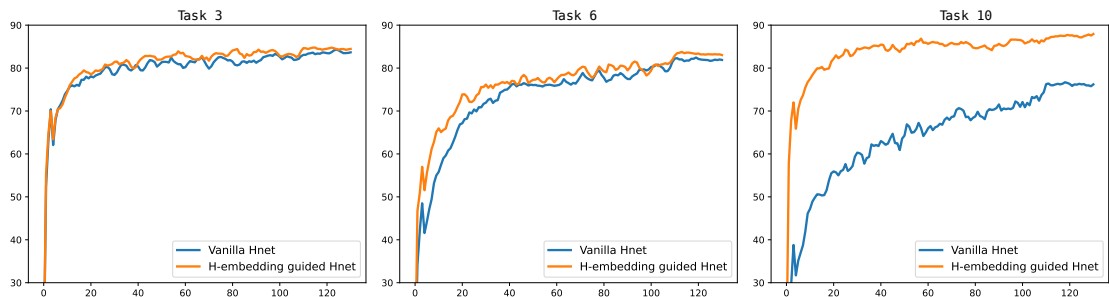

Figure 4: **Plotting of test accuracy during training task 3, 6, 10 of CIFAR-100,** with axis x and y for the number of checkpoints and accuracy respectively. The blue curve represents the vanilla hypernet and the orange represents our H-embedding guided hypernet. As CL progresses, our method exhibits quicker convergence to higher accuracy in later tasks.

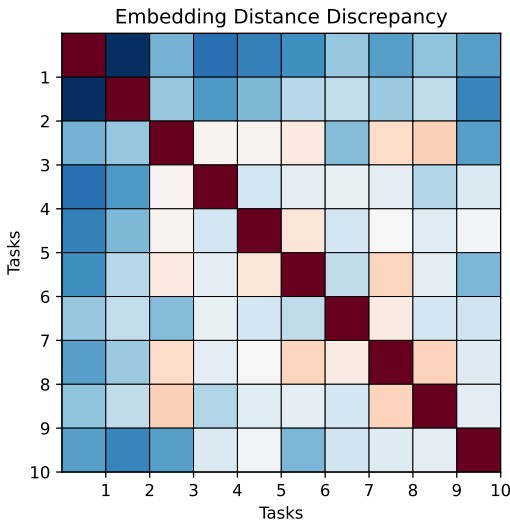

Figure 5: **Visualization of discrepancy between the task embedding distances learned w/ and w/o H-embedding guidance.** The grid of $i$-th row and $j$-th column represents the distance of task $i$ and $j$. Darker cells indicate a larger discrepancy, with red for d(w/) < d(w/o) and blue vice versa.

CIFAR-100 split task covering classes of people and reptiles, as an instance. The embedding derived in our H-embedding guided hypernet successfully marks tasks 3, 5, 6, 7, 9 as more related, which all contain coverage of terrestrial animal classes or human scenarios. Such correspondence with human intuition suggests a better capture of task interrelationships, leading to higher CL efficiency.

**Optimal Overall Transfer Ability.** Following Sec. 5.3, we assess the FWT and BWT performance of all methods in comparison study and list them in Table 7. Our framework derives the best overall ability, displaying competitive performance in both forward and backward transfer.

| Task | CIFAR-100 | | | ImageNet-R | | |
|---|---|---|---|---|---|---|
| **Method** | $\mathcal{FAA}$ (↑) | $\mathcal{BWT}$ (↑) | $\mathcal{FWT}$ (↑) | $\mathcal{FAA}$ (↑) | $\mathcal{BWT}$ (↑) | $\mathcal{FWT}$ (↑) |
| Finetune (ResNet) | 19.13 | -59.89 | 0.20 | 15.08 | -41.34 | 19.74 |
| LwF | 34.35 | -50.20 | 5.73 | 17.30 | -23.36 | 3.39 |
| EWC | 37.83 | -46.38 | 5.40 | 15.77 | -12.20 | -9.47 |
| L2 | 41.49 | -43.47 | 6.14 | 15.95 | -47.82 | 26.34 |
| PredKD + FeatKD | 34.43 | -49.29 | 4.91 | 18.22 | -22.15 | 3.39 |
| PackNet | 70.68 | - (N/A) | -8.14 | 34.63 | - (N/A) | 1.99 |
| HyperNet | 81.57 | -0.06 | 2.82 | 38.03 | -0.15 | 4.88 |
| WSN | 82.75 | - (N/A) | 3.94 | 37.99 | - (N/A) | 5.67 |
| H-embed Hnet* | **83.08** | -0.01 | 4.27 | **38.16** | 0.07 | 4.80 |
| L2P | 83.18 | -4.51 | 7.34 | 71.26 | -4.87 | 3.82 |
| DualPrompt | 81.48 | -4.93 | 6.06 | 68.22 | -5.59 | 1.50 |
| CODA-Prompt | 86.31 | -4.36 | 10.32 | 74.05 | -4.09 | 5.83 |
| HiDe-Prompt | 93.48 | -1.54 | 14.67 | 74.65 | -3.81 | 6.15 |
| InfLoRA | 86.75 | -4.97 | 11.37 | 74.75 | -5.92 | 8.36 |
| SD-LoRA | 87.26 | -4.79 | 11.70 | 77.18 | -4.56 | 9.43 |
| H-embed Hnet-LoRA* | **97.07** | 0.01 | 16.71 | **81.38** | -0.29 | 9.36 |

Table 7: **Accuracy (%) and Transfer (%) Comparison on CIFAR-100 and ImageNet-R.** All range of results are derived by three times running with different random seeds and calculating the average. Our method (marked by '*') achieves the top average accuracy with high confidence.

