# OpenReview forum: "Exploiting Task Relationships in Continual Learning via Transferability-Aware Task Embeddings"
_NeurIPS.cc/2025/Conference — NeurIPS 2025 poster_

### Official Review · Reviewer_ZN1R · 2025-06-10

**Clarity:** 3
**Significance:** 2
**Originality:** 2
**Rating:** 5
**Confidence:** 2

**Summary:**

This paper explores how the H-score can be leveraged to model task relationships in continual learning (CL). It introduces an online task embedding mechanism called H-embedding, which feeds into a hypernetwork. The hypernetwork, trained jointly with the embeddings, predicts the weights of the inference model directly from these embeddings. The method is evaluated on CL benchmarks across various vision architectures, including ResNets, ViTs, and ViTs+LoRA adapters.

**Questions:**

The paper presents ideas that are theoretically grounded, though at times they come across as informed guesses about what might work best. That said, the results are strong, and the paper is both technically solid and well-written.

I encourage the authors to answer the points presented in the Strengths/Weaknesses section of this review, particularly the concerns around understanding and controlling the behavior of the proposed framework. For now, I see no reason for a lower the score, and I’m open to revising my assessment after considering the authors' response and other reviewers’ comments.

**Ethical Concerns:**

["NO or VERY MINOR ethics concerns only"]

**Final Justification:**

All of my primary concerns have now been resolved. While I agree that some reviewer remarks deserve further discussion, they are minor and can be handled during the camera-ready stage. Therefore, I increased my score.

**Limitations:**

yes

**Quality:**

3

**Strengths And Weaknesses:**

The paper is formally sound and presents promising results. However, the contribution appears to rely primarily on combining existing components (i.e. adding modules and optimizing loss functions) without a clearly articulated breakthrough. While the empirical improvements are appreciated, the core mechanisms driving these gains remain somewhat opaque.

For instance,

- In the ablation study (Fig.3), it is unclear why removing the H-embedding is detrimental at $N=10$, but has negligible impact at $N=5$ and $N=20$.

- Similarly, the AHP component seems to offer little performance benefits, yet it is retained in the final framework without sufficient justification.

Also, to better understand the contribution of your work, I would appreciate clarifications on the following points:

- The paper seemingly is missing pointers clearly stating from which papers the benchmarks are inherited from (i.e., the baselines are all cited, but it could be useful to also understand if the specific dataset-architecture pairs were already studied in the literature, or these are novel test beds the authors are proposing)

- The authors mention re-running some methods while transcribing results of others from the literature. Could the authors clarify which results were reproduced versus taken directly?

- Have the authors considered comparing methods in terms of inference-time efficiency (both memory usage and runtime)? This would provide a full picture of the practical trade-offs to justify this added complexity.

- What are the computational costs (in both memory and time) of training the Hypernetwork component?

---

> ### Author Rebuttal · Authors · 2025-07-31
>
> We are grateful for the reviewer’s encouraging comments and thoughtful suggestions, and we address the raised points in detail below.
> ## (Weakness 1)
>
> Thank you for your positive comments on the soundness and empirical results of our paper. We would like to further clarify the novelty and motivation behind our method. Although our framework involves known components (e.g., the general architecture of a hypernetwork for parameter generation), it introduces key conceptual and technical innovations that set it apart from prior works. As stated in the Introduction, the key insight of our work is that task relationship can provide prior information of the task space structure in the continual learning context. Upon reviewing prior literature, we observed that most existing strategies are primarily performance-driven and heavily rely on training procedures, with limited consideration of prior inter-task information. Motivated by this gap, we proposed the use of task embeddings to guide the learning process, and designed a transferability-aware task embedding that is compatible with continual learning constraints—computed without accessing past data or models—thereby providing a principled way to inject task structure priors into the training process. Specifically, on the basis of the transferability metric H-score, we 1) proposed to use HGR divergence as a unified definition of self transferability conforming to the H-score theory framework; 2) introduced AHP normalization for transferability scale alignment across different tasks; 3) defined an optimization problem to distill the transferability information into a low-dimensional task embedding，enabling the incorporation of task-level information into continual learning strategies.
>
> Moreover, our use of existing components is not a simple reuse or combination; rather, we carefully adapt and integrate them into our framework to serve our specific goals. For instance, compared to conventional applications of HyperNetworks in continual learning, we reformulated the HyperNet pipeline to improve its computational efficiency and enabled it to generate only LoRA parameters. These modifications substantially reduce both training and inference overhead, which is particularly valuable in modern continual learning scenarios that demand lightweight adaptation, setting our approach apart from most previous uses of HyperNetworks, which typically generate full model weights and thus incur significantly higher storage and compute costs.
>
> As for the listed concerns, we address them below:
>
>  - **Explanation for Fig. 3a Performance Variation and Impact of Introducing H-Embedding:**
>
>    In the current version, Fig. 3a presents single-seed results. The performance drop of FAA “w/o Hemb” on the ImageNet-R 10-task benchmark suggests a degree of robustness issue with vanilla HyperNet when generating LoRA parameters across different random seeds, which also appears on 5-task and 20-task benchmarks. To provide a more stable and comprehensive comparison, we additionally report multi-seed results of all three task settings in the Table, including mean and standard deviation. It can be observed that introducing our task-guided embeddings helps stabilize the performance and improves the overall average. We plan to further investigate the underlying cause of this phenomenon in the future.
>
> | Benchmarks    | Vanilla Hypernet (%)  |HyperNet with H-Embedding Guidance (%) |Increasing Ratio|
> |----------------|------|-----------|---------|
> |ImageNet-R 5 task | 74.09 (10.99)  | 78.29 (0.43)| 5.66% |
> |ImageNet-R 10 task |   76.77 (10.19)  | 80.89 (1.12)  | 5.36%  |
> |ImageNet-R 20 task |74.98 (5.78)   |  80.41 (2.24) | 7.24% |
>
>    Regarding the significance of performance improvement, in average our improvements compared to removing H-embedding are notable and consistent. For example, in ImageNet-R benchmark, recent state-of-the-art method (SD-LoRA) improves upon hypernet by around 1.95%, while our method achieves an additional 4.06% improvement over it and 6.09% over the original HyperNet (measured by averaged performance across 7 seeds and 3 task splits). Also, in our experimental studies, the performance gain of our framework is consistently observed across different backbones (e.g., CNN, ResNet, ViT) and benchmarks (e.g., CIFAR100, ImageNet-R, DomainNet), suggesting that our task-relation-guided prior offers meaningful benefit.
>
>    We will clarify these points in the revised paper and include modify Fig. 3a to include multi-seed results for to better illustrate robustness.
>
>  - **AHP normalization:**
>
>    As mentioned in our response regarding the design of H-embedding, the introduction of AHP is primarily motivated by the need to perform transferability scale alignment across different tasks, which we believe is an intuitive and interpretable design. This design helps prevent scale mismatch with the embedding distances and ensure that the transferability signals remain comparable across tasks when used as guidance.
>
>    In essence, AHP serves as a normalization post-processing step for the transferability values, rather than introducing new information. This may partly explain its relatively modest quantitative impact. Nevertheless, incorporating AHP still leads to an empirical performance gain of approximately 4% (averaged across 3 task splits). While this improvement is not decisive, it demonstrates that AHP provides a non-trivial contribution to the final performance and justifies its inclusion in the framework.
>
> ## (Weakness 2)
>
> Regarding the experimental setup, we would like to clarify that all our experimental benchmarks—both the datasets and their specific splits—are directly adopted from existing works that are well-established and repeatedly used in the recent literature (e.g. [Liang and Li, 2024], [Wu et al., 2025]). The architectures we employ, ViT-B/16 and ResNet-32, are also widely used and considered representative choices in prior studies. Therefore, we do not propose any new test beds in this work; this design choice was intentional to ensure that comparisons with previous methods are fair and objective.
>
> ## (Weakness 3)
>
> The baselines for full-parameter ResNet experiments (including CIFAR-100 and ImageNet-R benchmarks) were reproduced by us, while the baselines for ViT-based parameter-efficient fine-tuning (PEFT) were cited from existing literature. In particular, we mainly cited results from the SD-LoRA [Wu et al., 2025] paper (including L2P, DualPrompt, CODA-Prompt, HiDe-Prompt, InfLoRA, and SD-LoRA), and the results of HiDe-Prompt on CIFAR-100 benchmark was taken from its original paper.
>
> This choice was made because the full-parameter methods span a relatively long time range, and many of them do not share a unified benchmark setting or backbone architecture. To ensure fairness and consistency, we reproduced all these baselines under our evaluation framework. In contrast, most of the ViT-PEFT methods are recent works with more consistent experimental setups, making it more reasonable to cite their results directly.
>
> ## (Weakness 4)
>
> Despite its seemingly complexity, the hypernet framework does not incur significant time costs or memory usage during inference, as the number of parameters in the hypernetwork has been strictly controlled. In the original hypernet framework, the number of hypernet parameters is roughly comparable (or smaller) to that of the main network. Therefore, if processing one sample at a time, a single prediction involves one forward pass through the hypernet and one through the main net, theoretically leading to an inference time approximately twice that of a regular model. However, in practice, we only need to perform one forward pass through the hypernet per task, which means that the amortized time cost per sample becomes negligible when the dataset is large. The situation is even more favorable in the ViT-LoRA setting, where the hypernet is only responsible for generating LoRA parameters. Since these modules are much smaller in size than the full ViT model, the forward computation of the hypernet contributes only marginally to the overall inference time. We conducted runtime tests for both the ResNet and ViT backbones using full testing set, and the results support our analysis.
>
> | Backbone & Benchmark | Time Cost of Plain Backbone Model (s) | Time Cost of H-Embedding Guided Hypernet Framework (s) | Parameter Size of Plain Backbone Model |Parameter Size of H-Embedding Guided Hypernet Framework |
> |----------|---|---|---|---|
> |ResNet on Cifar100|4.257|4.260| 468540 | 457336 |
> |ViT on ImageNet-R |4.997| 5.041| 743624(LoRA) | 458424 |
>
> We believe that the comparison of inference time and parameter size between our method and the plain backbone already demonstrates the efficiency of our framework at inference time. Due to time constraints in the rebuttal phase, we were unable to conduct a comprehensive comparison with all baseline methods, but we hope the above analysis and measurements sufficiently support the claim that our method introduces minimal inference overhead.
>
> ## (Weakness 5)
>
> For similar reasons as stated in analysing the inference efficiency of our framework, the training computational costs of training hypernet is also relatively minor. We list the results of 1-epoch runtime tests of ViT-LoRA backbone on ImageNet-R 10 task benchmark below:
>
> | Task ID    | 1  | 2 | 3| 4|5|6|7|8|9|10|
> |----------------|------|-----------|---------|---|---|---|---|---|---|---|
> | HyperNet Total (s) |  4.68 | 5.67 | 5.02 |6.24|5.67|6.18|5.36|6.26|5.63|5.46|
> | LoRA Finetune Total (s)  | 4.37| 5.16|4.41|5.40|4.90|5.17|4.41|4.94|4.41|4.15|
>
> As shown, the 1-epoch training time of our method is slightly higher than that of plain LoRA, but the difference is minor and acceptable. Given their similar parameter sizes, the memory usage is also within a reasonable range.

---

> ### Comment · Reviewer_ZN1R · 2025-08-02
>
> I sincerely thank the authors for their detailed and thoughtful rebuttal. After carefully reading the responses, as well as the comments from the other reviewers, I believe the authors have addressed the majority of the concerns raised during the initial round of review.
>
> That said, I would like to raise a few remaining points that I believe, if addressed in the camera-ready version, would further improve the clarity and completeness of the work:
>
> ---------------
> **1.** The use of H-score-based embeddings is motivated. However, for the benefit of readers, I would encourage the authors to either:
> - Include a brief ablation study substituting H-score with an existing embedding method (eg. Task2Vec [1] or LEEP [2], see **Reviewer JmhA** comments), or
> - If such experiments are not feasible at this stage, add a concise discussion explaining why these alternatives may not be suitable for your setting.
>
> ---------------
> **2.** While the authors have shown that the training and inference time overhead is relatively minor, it would be valuable to also report peak GPU memory consumption during both training and inference, so to have a more complete view of the method's requirements.
>
> ---------------
>
> If the authors are able to incorporate these remaining points into the final version, I would be inclined to raise my score.
>
> ---------------
> **__References:__**
>
> [1] Achille, Alessandro, et al. "Task2vec: Task embedding for meta-learning." Proceedings of the IEEE/CVF international conference on computer vision. 2019.
>
> [2] Nguyen, Cuong, et al. "Leep: A new measure to evaluate transferability of learned representations." International Conference on Machine Learning. PMLR, 2020.

---

> ### Author Response · Authors · 2025-08-07
>
> We sincerely thank the reviewer for the thoughtful follow-up and the constructive suggestions. We are glad that the core concerns have been addressed and appreciate the opportunity to further improve the clarity of our work. We address each of the two remaining points below:
>
> **(1) On the use of H-score versus other task embedding methods:**
>
> Regarding the suggestion to consider alternative task embedding strategies, we appreciate this insightful point. We would first like to clarify the rationale behind our proposed H-embedding and the use of H-score within our framework. While technically task embeddings can be estimated using various methods, the alternatives are generally less suitable for our target setting, as they either require access to source data for each task—assumptions that are infeasible under rehearsal-free CL—or do not explicitly model task-to-task relationships, thereby offering limited benefit in facilitating bidirectional transfer across CL tasks.
>
> To address this, we propose a new formulation where task embeddings are computed online by explicitly optimizing them to approximate a transferability metric across tasks. In this design, we adopt H-score as the target transferability metric primarily due to its source-data-free nature, which allows us to maintain compliance with CL constraints. Additionally, H-score is theoretically grounded, computationally efficient, and empirically shown to correlate well with actual transfer performance. These characteristics make it a particularly suitable and representative choice for our embedding algorithm. Yet, our framework is not inherently tied to H-score, and there admittedly exist other source-free embeddings or transferability metrics (e.g. task2vec, LEEP, LogME) that could also, in principle, be used within it.
>
> As both an ablation and a further assessment of our framework's flexibility, we conducted preliminary experiments on ImageNet-R 10 task benchmark by:
>
>  - Replacing H-score with other source-free transferability metrics (LEEP and LogME) while keeping the embedding algorithm and overall pipeline unchanged;
>  - Replacing the entire H-embedding formulation with Task2Vec.
>
> The results are shown below:
>
> |HyperNet with H-Embedding (%) | HyperNet with H-embedding-LEEP (%) |  HyperNet with H-embedding-LogME (%) |Hypernet with task2vec (%) |
> |-----------|---------|--|--|
> | 81.8 | 81.6 |  81.4 | 79.8 |
>
> As shown, replacing the transferability metric alone leads to only minor drop in performance. This supports our view that while H-score is a strong instantiation, the core contribution of our work lies in the transferability-guided embedding formulation and its integration into a hypernetwork-based parameter generation pipeline—a general framework that remains valid even when alternative metrics are employed.
>
> However, replacing the entire embedding formulation with task2vec leads to a more noticeable performance degradation. We believe this is because Task2Vec computes task embeddings independently without explicitly modeling task-to-task relationships, which limits its capacity to guide transfer dynamics in CL. Additionally, Task2Vec is highly sensitive to the choice and training of the probe network, which introduces extra variance in effectiveness and incurs additional memory costs—challenges not present in our H-embedding approach.
>
> Due to the limited time available during the rebuttal phase, we focused on a representative subset of alternatives rather than attempting an exhaustive evaluation. We will incorporate the above results into the final version along with a more comprehensive discussion. Further exploration of other embedding designs also remains an important direction for our future research.
>
> **(2) On memory usage:**
>
> We agree that this is an important consideration. In our internal profiling, we observed that the peak GPU memory usage of our method is slightly lower than that of the vanilla ViT backbone during inference (ours: 5808 MiB vs. ViT-LoRA original: 5832 MiB), but approximately twice as high during training (ours: 24,970 MiB vs. ViT-LoRA original: 13,022 MiB). These values are rough measurements based on a fixed batch size of 128 on ImageNet-R using NVIDIA A800 GPUs. Due to time constraints, we have not yet profiled other datasets or configurations. We note that actual GPU memory consumption can vary depending on batch size, PyTorch's memory allocation behavior, and implementation details. The higher training-time usage in our method is likely caused by additional memory demands in loss computation and backpropagation, which we have not specifically optimized. Since memory was not a primary bottleneck in our experiments, we did not focus on its minimization. Nevertheless, we recognize the value of reducing memory overhead, and are open to explorations of more efficient engineering solutions in the future.

---

> ### Comment · Reviewer_ZN1R · 2025-08-07
>
> I thank the authors for their comprehensive and timely response. All of my primary concerns have now been satisfactorily addressed. Accordingly, I have increased my score.

---

> > ### Author Response · Authors · 2025-08-09
> >
> > Thank you very much for your time and expertise. We sincerely appreciate your constructive feedback and positive assessment of our work. Your comments have helped us a lot in refining our paper, as well as greatly encouraged us to continue advancing this research direction.

---

### Official Review · Reviewer_Wucx · 2025-06-28

**Clarity:** 3
**Significance:** 3
**Originality:** 3
**Rating:** 4
**Confidence:** 4

**Summary:**

The H score based on model parameters from multiple tasks are used to learn a sequence of embedding vectors, and the similarity of these embedding vectors is used to guide task relatedness of a hypernet network for design of the parameters for the next task. Since the hypernet parameters are updated with each new task, the model is forward and backward information flow as a function of sequential tasks. There are multiple elements to the model (it is fairly complicated, with many networks interacting), including an encoder-decoder linking model features to the aforementioned embeddings. A comprehensive set of experiments are presented, including ablation studies.

**Questions:**

The H-score is a key element of this model. I think the definition in (2) should be clarified. You write "cov" in two places, and these mean different things. I think (2) is unnecessarily ambiguous. I had to go read Huang et al (2019) to understand it, and I think they gave a clearer definition. Please make clearer in a revision.

How well is this model going to scale for large \theta^{(n)}? It seems hard for the Hypernet to work well for very large models.

There are a lot of "moving parts" to this model. In (9) there is \beta_e and \beta_c, and then from Sec 4.2.2 there is encoder-decoder. It would seem that this model could get into local modes, possibly poor ones, given the complexity of this model, particularly for large \theta^{(n)}. Also, the model could be very dependent on the dimension of e^{(j)} and its characteristics. Please comment.

**Ethical Concerns:**

["NO or VERY MINOR ethics concerns only"]

**Paper Formatting Concerns:**

No issues.

**Quality:**

3

**Strengths And Weaknesses:**

Strengths:

Overall, the paper is well written, but there are some elements that could be clarified (questions below).
The experiments are strong.

Weaknesses:

While the model makes sense, it is quite complicated. It is not clear how well this will scale to very large models, and large \theta. Additionally, key to the whole thing is the dependence of the hypernet on e^{(j)} and the ability of the model to infer task interrelatedness, via the H-score. How sensitive are the results to the dimension of e^{(j)}? Since e^{(j)} plays such an important role, it seems its characteristics are important. This seems to be why so much regularization is done, for example via the encoder-decoder.

---

> ### Author Rebuttal · Authors · 2025-07-31
>
> We sincerely appreciate the reviewer’s positive evaluation and the insightful feedback provided on our work. Below, we address each of the points in detail.
>
> ## (Weakness 1 & Question 2)
>
> We agree that scaling full-parameter generation via hypernetworks to very large models can raise efficiency concerns. To address this, one of the key contributions of our work is to combine current hypernet frameworks with parameter-efficient fine-tuning (PEFT) techniques. Specifically, we employ the hypernet to generate LoRA parameters instead of full weights in our current design, with potential for future extensions to other module types. This design enables our method to be flexibly integrated into a wide range of large-scale models as a plug-in module, allowing us to leverage the capabilities of powerful pretrained backbones while keeping the hypernetwork lightweight and efficient.
> As one of our main comparison studies, the HyperNet-LoRA framework has demonstrated strong and stable performance across our experiments, confirming the effectiveness of this design.
>
> ## (Weakness 2)
>
> The learned embedding indeed plays a critical role in hypernetwork-based frameworks, which is also one of the reason we propose to guide its learning through a transferability-aware objective. Yet, in our framework, the performance is relatively insensitive to the embedding dimensionality as a hyperparameter. To support this, we conducted experiments with embedding dimensions set to 8, 16, 24, and 32 on ImageNet-R 10 task ViT setting. As shown in the table below (`*' denotes the setting in experimental results), although FAA exhibits slight fluctuations across settings, the overall performance remains stable. This suggests that our method is relatively robust to changes in embedding size, and does not rely on fine-tuning this hyperparameter to achieve good results.
>
> | Embedding Size    | FAA |
> |----------------|------|
> |8 | 80.245 |
> |16 | 81.777  |
> |24 |   80.996  |
> |32*| 80.970    |
>
> We appreciate the reviewer’s insightful comment regarding the characteristics of the embedding. We agree that this aspect is indeed worth further investigation, as it could provide valuable insights into the behavior of transferability-aware hypernetworks. We hope to conduct a more comprehensive study in this direction in future work.
>
> ## (Question 1)
>
> We thank the reviewer for the valuable feedback and agree that the definition in Eq. (2) can be clarified further. In future revisions, we will provide a more intuitive explanation of the formula to enhance its readability. Although the two instances of "cov" might suggest different interpretations, they are consistent from a computational standpoint — both denote the covariance computed over the first dimension (i.e., across the observations) of the input data. We will explicitly clarify this point in the revised manuscript.
>
> ## (Question 3)
>
> We appreciate the reviewer’s thoughtful observation. Firstly, we would like to clarify that compared to typical deep learning methods, our framework does not introduce significantly more “moving parts”. The two hyperparameters $\beta_e$ and $\beta_c$ serve as standard loss balancing factors, which are commonly adopted in many existing works; the encoder and decoder modules are designed as very shallow linear networks, jointly trained with the main HyperNet, making the overall architecture relatively straightforward and within the scope of typical deep learning designs.
>
> To further illustrate this, we provide a parameter count comparison showing that the total parameter size of our HyperNet framework is comparable to that of conventional model architectures. The connections involved are also quite simple, and the training is end-to-end. So far, we have not observed any indications of local minima or poor local optima issues during experiments. Regarding the sensitivity to the hyperparameters $\beta_e$ and $\beta_c$, our empirical results suggest that the framework is relatively robust to their variations. Below we summarize the performance under different settings of these hyperparameters on ImageNet-R 10 task ViT setting (`*' denotes the setting in experimental results).
>
> |beta_c \ beta_e | 0.01| 0.03|0.05* |0.07|
> |---|---|---|---|---|
> |0.01 |0.8188 |0.8169| 0.8117 |0.8132|
> |0.03| 0.7686| 0.7788 |0.7771 |0.7850|
> |0.05*| 0.7851| 0.7755 |0.7892 |0.7850|
> |0.07| 0.7763| 0.7882 |0.7852 |0.7878|

---

> > ### Comment · Reviewer_Wucx · 2025-08-05
> >
> > Thank you for your thoughtful response. I will leave my already-high score as is.

---

> ### Author Response · Authors · 2025-08-09
>
> We sincerely thank you for your helpful comments and positive evaluation of our paper. We will continue to refine our work based on your valuable suggestions.

---

### Official Review · Reviewer_JmhA · 2025-07-02

**Clarity:** 3
**Significance:** 2
**Originality:** 3
**Rating:** 4
**Confidence:** 5

**Summary:**

The paper proposes a continual learning framework that introduces a novel transferability-aware task embedding, H-embedding, derived from an information-theoretic H-score metric to capture inter-task relationships. This embedding guides a hypernetwork to generate task-conditioned weights, achieving improved forward and backward transfer on several CL benchmarks without rehearsal.

**Questions:**

- Please address the questions in Weaknesses.
- In Fig. 3a, why is the FAA performance of "w/o Hemb" poor for ImageNet-R 10 Tasks, but competitive for ImageNet-R 5 Tasks and 20 Tasks?
- In Fig. 3b, why does the proposed method achieve the same DAA for both TIL and CIL versions?
- an the H-embedding be replaced by other task embeddings, such as those in Task2Vec (Achille et al., TASK2VEC: Task Embedding for Meta-Learning. ICCV’19) or by embeddings computed with different transferability metrics? How would this choice affect performance?
- For ViT-LoRA, is the H-score computed using the CLS token features?

**Ethical Concerns:**

["NO or VERY MINOR ethics concerns only"]

**Final Justification:**

The authors have addressed my main concerns. Therefore, I am increasing my score from 3 to 4.

**Limitations:**

yes

**Paper Formatting Concerns:**

no concerns

**Quality:**

3

**Strengths And Weaknesses:**

Strengths:
- The proposed framework is storage-efficient, requiring storing only low-dimensional task embeddings.
- The proposed framework is compatible with parameter-efficient finetuning (PEFT) settings like LoRA.
- The paper presents extensive empirical comparisons against strong baselines across multiple datasets (CIFAR-100, ImageNet-R, DomainNet) and backbones (ResNet, ViT), showing improvements in both FAA and DAA.

Weaknesses:
- The contribution is not very significant compared to HyperNet \[von Oswald et al., 2020]. The empirical gain over HyperNet appears notable only for FAA on ImageNet-R 10 Tasks with ViT-LoRA, suggesting that the benefit from the task relation prior is limited.
- Methodological concerns:
	- In Eq. (11), both the encoder and decoder are trainable and can be updated to produce outputs close to \$\hat{e}^{(j)}\$, but there is no constraint ensuring these outputs are close to the true \$e^{(j)}\$. How does this setup ensure that \$e^{(j)}\$ aligns well with \$\hat{e}^{(j)}\$?
	- Lines 168–170: The AHP-normalized transferabilities are converted to distances using a “standard affinity–distance” method. However, the given formula for \$dist(T\_n, T\_j)\$ is usually used to convert distance to affinity, not the other way around. Also, the motivation for introducing the additional learnable scale \$\gamma^{(j)}\$ is unclear.
	- It is not clear why transferability needs to align with the symmetry property of Euclidean distance.
- The paper lacks discussion and comparison with related work that also uses transferability to facilitate continual learning: Ermis et al. Memory Efficient Continual Learning with Transformers. NeurIPS'22.
- High training cost: When the number of tasks is large, the method needs to generate task-specific networks, use them to compute H-scores, and calculate continual learning losses. This can significantly increase computational cost and training time.
- Limited extension to class-incremental learning (CIL): The paper only briefly mentions using an auxiliary task ID classifier on pre-trained image features. This solution is not well integrated with the main framework, which the paper also acknowledges as a limitation.

---

> ### Author Rebuttal · Authors · 2025-07-31
>
> We sincerely thank the reviewer for the thoughtful and detailed feedback. Below, we respond point-by-point.
>
> ## (Weakness 1 & Q1)
>
> For the concerns about HyperNet [von Oswald et al., 2020] comparison and the performance fluctuations shown in Fig. 3a, we address both problems below.
>
>  - **Performance Significance Compared to HyperNet:**
>     Despite its limitations, we acknowledge that the original HyperNet is a relatively mature CL strategy, and surpassing it by a large margin is naturally more challenging than typical ablation settings. However, we believe our improvements over HyperNet as a baseline are still notable and consistent. For example in ImageNet-R benchmark, recent SOTA method (SD-LoRA) improves upon hypernet by 1.95%, while our method achieves a 4.06% improvement over SD-LoRA and 6.09% over the original HyperNet (measured by averaging across 7 seeds and 3 task splits). Also, in our experimental studies, the performance gain of our framework is consistently observed across different backbones (e.g., CNN, ResNet, ViT) and benchmarks (e.g., CIFAR100, ImageNet-R, DomainNet), suggesting that our framework offers meaningful benefit beyond HyperNet.
>
>  - **Explanation for Fig. 3a Performance:**
>     Currently, Fig. 3a presents single-seed results. The performance drop of “w/o Hemb” on the ImageNet-R 10-task benchmark suggests a degree of robustness issue with vanilla HyperNet when generating LoRA parameters across different random seeds, which also appears on 5- and 20-task benchmarks. To provide a more stable and comprehensive comparison, we additionally report multi-seed (7 in total) results of all three task settings in the Table, including mean and standard deviation. It can be observed that introducing our task-guided embeddings helps stabilize the performance and improves the overall average. We plan to further investigate the underlying cause of this phenomenon in future.
>
> | #Tasks    | Vanilla Hypernet (%)  |HyperNet w/ Hemb (%) |Increasing Ratio|
> |---|--|---|---|
> |5 | 74.09 (10.99)  | 78.29 (0.43)| 5.66% |
> |10 |   76.77 (10.19)  | 80.89 (1.12)  | 5.36%  |
> |20 |74.98 (5.78)   |  80.41 (2.24) | 7.24% |
>
>  - **Novelty and Methodological Contributions:**
>     While our framework adopts a HyperNet-like architecture for parameter generation, it substantially differs from the previous work in both motivation and implementation. The central contribution of our work lies in the introduction of task relation priors, which guide the dynamic adaptation of model parameters across tasks. In contrast to the vanilla hypernet, where task embeddings are learned implicitly through task-specific data, we explicitly model and compute inter-task affinities to guide the learning of task embeddings and hypernet, resulting in better generalization and stability across continual tasks.
>
>     Moreover, as part of our contribution, we reformulated the implementation of HyperNet pipeline, improving its computational efficiency and enabling it to generate only LoRA parameters. These adjustments significantly reduce both training and inference overhead, and are especially beneficial in modern CL scenarios where lightweight adaptation is critical. It also distinguishes our work from most prior uses of HyperNet, which typically generate full model parameters or require heavier storage and compute budgets.
>
>   We will clarify these points in the revised paper with Fig. 3a including multi-seed results.
>
> ## (Weakness 2)
>
> ### 2.1
>
> Eq.(11) is not intended to align the additional output with $\hat{𝑒}^𝑗$,  but rather to enforce alignment between $\hat{𝑒}^𝑗$ and the transformed version of $e^j$ through the encoder-decoder mapping. In essence, this alignment ensures that the guiding task embedding $\hat{𝑒}^𝑗$ can be well approximated by a learned transformation of $e^j$, or intuitively from an information transmission perspective, $e^j$ should retain sufficient information to recover the H-embedding $\hat{𝑒}^𝑗$.
>
> The encoder-decoder structure can be equivalently viewed as an autoencoder, of which the hidden layer is coupled with the intermediate layer of the hyperNet, facilitating joint optimization and promoting a consistent representation space between embeddings and hyperNet outputs. This type of alignment using autoencoders has been widely adopted in previous literature(e.g., [Schonfeld et al., 2019]). It does not require reconstruction of the original $e^j$; rather, the core idea is to provide a bridge between the prior calculated H-embedding and the embedding being learned. We described it as an encoder-decoder module to maintain conceptual clarity and avoid confusion with the core HyperNet mechanism.
>
> ### 2.2
>
> We agree that the more common formulation is: $aff = exp(-\gamma \times dist)$, yet we adopts the reverse mapping: $dist = exp(-\gamma \times aff)$. Nevertheless, this reverse form is not incorrect, and has been used in prior works too — notably in Taskonomy [Zamir et al., 2018]. Intuitively, both affinity and distance are relational measures where the relative scale matters more than the absolute value. Such a transformation is to ensure a smooth inverse relationship, and our formulation also satisfies this requirement.
>
> The inclusion of the scaling factor $\gamma$ further enhances this effect: it adjusts the steepness of the exponential curve, ensuring the learned embedding distance emphasizes relative differences without being dominated by absolute magnitude. This is especially important in our framework where the scale sensitivity of H-score transferability across different target tasks can affect the optimization stability of H-embedding computation.
>
> ### 2.3
>
> We acknowledge that transferability is inherently asymmetric, and does not naturally match the symmetry property of Euclidean distance. However, in our framework, this alignment remains a reasonable and effective approximation, due to the following:
>
>  - **Asymmetric guidance is sufficient:** In CL, we are primarily concerned with how previous tasks affect the current task. Thus, we only require one-way transferability and inter-task relation, and symmetric alignment is not strictly necessary.
>
>  - **Constraint of CL:** Since our framework operates in a strict rehearsal-free CL setting, we cannot revisit or recompute full two-way transferability after a task is finished. The one-way transferability from known tasks is the only feasible choice.
>
> Overall, we believe this approximation still provides meaningful structure for guiding embedding space learning and has been validated by empirical improvements.
>
> ## (Weakness3 & Q3)
>
> In this work, we mainly focus on the increasingly popular rehearsal-free CL setting, where access to past data is prohibited. In contrast, [Ermis et al., 2022] relies on storing both past models and selected samples, which violates our problem setting. Therefore, this work is not currently included in our main comparisons. Nevertheless, we still thank the reviewer for bringing it up and the underlying ideas are indeed related to ours. We'll include a discussion in the revised version.
>
> Despite the setting mismatch, we conducted an additional experiment. As the code for the paper is not publicly available and the full reproduction would require considerable effort, we instead implemented our method on the benchmark (CIFAR-100 20 tasks + ViT) used in Fig 3 of the original paper, with their method reporting a FAA of approximately 0.78 and our method achieving 0.907.
>
> Since the original paper does not provide detailed numerical results, we estimated their performance by reading values from the figure. Also, due to differences in dataset configurations, the current comparison may not be entirely fair. We will conduct a more rigorous and fair comparison once we have sufficient time and resources in the future.
>
> This also partially addresses Q3. Our choice of metric was driven by practical considerations. In rehearsal-free CL, it is infeasible to store large amounts of past data or model parameters between tasks. The H-score metric stands out because it requires no source data, and the only information it needs—source model parameters—can be derived using the HyperNet from the task embedding. Hence, our framework strictly adheres to the CL constraints, while still enabling fast and theoretically grounded estimation of task relationship. Additionally, H-score is one of the most currently recognized transferability metrics in the field, known for its solid theoretical backing and strong empirical correlation with true transfer performance. We believe these make it a well-justified choice.
>
> We acknowledge that our current study does not cover the full spectrum of possible solutions. However, the presented method offers a feasible and effective approach to enhancing CL. Moving forward, we plan to explore alternative designs and conduct a more comprehensive analysis, in terms of both accuracy&efficiency.
>
> ## (Weakness 4)
> ...

---

> > ### Author Response · Authors · 2025-08-01
> >
> > Thank you again for the detailed comments. Due to the rebuttal word limit, we were unable to address all the points in the main response and therefore continue our reply here. We hope this is acceptable and appreciate your understanding.
> >
> > ## (W4)
> > It is admittable that computing H-embeddings introduces some additional cost. However, we'd like to clarify this overhead is relatively limited:
> >
> >  - The H-embedding is only computed once at the beginning of training for each task. This is a one-time initialization step and does not contribute to the per-epoch training cost, which dominates the overall runtime.
> >   - Computing H-embedding involves getting parameters for previous $j-1$ tasks via the hypernet, and using these models on current task $j$ to extract features. Since the HyperNet is no larger than the main network, this roughly amounts to $2(j-1)$ of inference time. Given the task number $j$ is typically small, the time overhead remains modest.
> >
> > To provide a concrete sense of the overhead, we measured the 1-epoch time cost on ImageNet-R 10-task benchmark. The results are summarized below:
> >
> > |Task ID|1|2|3|4|5|6|7|8|9|10|
> > |-|-|-|-|-|-|-|-|-|-|-|
> > |w/ Hemb Total(s)| 4.96|5.68|12.70|14.76|15.31|16.85|17.04|18.92|19.38|20.14|
> > |w/o Hemb Total (s)|4.68|5.67|5.02|6.24|5.67|6.18|5.36|6.26|5.63|5.46|
> > |Hemb Calculation (s)|-|-|7.90|8.29|9.46|10.72| 11.75|12.76|13.65|14.02|
> >
> > As shown, although the embedding calculation time increases with the task index, the growth is roughly linear and modest. The total time for computing the per-task H-embedding is approximately equivalent to 1~3 training epochs, while each task is typically trained for 50–100 epochs. Therefore, this additional cost is acceptable and does not significantly affect the overall efficiency.
> >
> > ## (W5)
> > We acknowledge this limitation. However, our primary contribution lies in introducing task relationship modeling as a guiding signal to improve learning dynamics in CL. While the current extension to CIL is relatively limited, it serves as a preliminary demonstration of our framework's potential applicability to CIL. Given the scope and focus of this work, we believe this provisional solution is a reasonable starting point. We see this not as a fundamental flaw, but as an exciting direction to expand upon. In future work, we'll develop more principled and integrated approaches tailored for CIL scenarios, further extending the utility of our framework.
> >
> > ## (Q2)
> > This is because the key difference between TIL and CIL lies in whether task identity is accessible during inference, while both is allowed access to task-specific information during training. Our method handles CIL by adding a task identity prediction module without changing training dynamics, so DAA remains unchanged across both settings, and only FAA reflects the difference.
> >
> > ## (Q4)
> > Currently, the H-score is computed using the average of the image patch features (i.e., excluding the CLS token), as the CLS token is often more sensitive to task-specific variations and less stable in scenarios involving task shifts.

---

> > ### Comment · Reviewer_JmhA · 2025-08-05
> >
> > Thank you to the authors for addressing most of my questions. However, several concerns remain:
> >
> > 1. For the response to Q3:
> > While the use of H-score-based embeddings is motivated, task embedding like Task2Vec and transferability like LogMe also do not require storing data or parameters from previous tasks. Ablation studies with these methods are important to clarify the advantages of using H-score-based embeddings.
> >
> > 2. For response to W2.3:
> > My original question was: Why does transferability need to align with the symmetry property of Euclidean distance? This question is raised due to the motivation for introducing AHP (lines 159-162) is unclear, especially since your response now says symmetry is not necessary.
> >
> > 3. For response to W2.2:
> > - In your paper, you use $dist = \gamma \exp(-aff)$, but in your response, you mention $dist = \exp(-\gamma \ aff)$. Please clarify which formula is actually implemented.
> > - I checked Taskonomy [Zamir et al., 2018] and did not find this specific affinity-to-distance formula.
> > - This conversion is not a standard affinity-distance method. Please avoid claiming otherwise in the paper.
> > - While I agree that relative scale is more important than absolute values, your affinity-to-distance conversion seems unintuitive. Let's ignore $\gamma$ here for simplification. Mapping distance to affinity as $\exp(-distance)$ is common, but the reverse (from affinity $[0,1]$ to distance $[\exp(-1), 1]$) narrows the range and reduces sensitivity to high affinity, which may not be desirable.
> >
> >
> > 4. For response to W2.1:
> > The full reference of [Schonfeld et al., 2019] is not provided.

---

> > > ### Author Response · Authors · 2025-08-08
> > >
> > > We are thankful to the reviewer for the detailed follow-up and questions. We address each remaining concern as follows:
> > >
> > > **(1) Regarding Q3 (alternative embeddings and ablations):**
> > >
> > > We appreciate the suggestion to include additional ablation studies. Yet, we would firstly like to clarify the rationale behind our proposed H-embedding and the use of H-score within our framework. While technically task embeddings can be estimated using various methods, the alternatives are generally less suitable for our target setting, as they either require access to source data for each task—assumptions that are infeasible under rehearsal-free CL—or do not explicitly model task-to-task relationships, thereby offering limited benefit in facilitating bidirectional transfer across CL tasks. To address this, we propose a new formulation where task embeddings are computed online by explicitly optimizing them to approximate a transferability metric across tasks. Despite the well supported choice of H-score as the transferability metric in our work, our framework is not inherently tied to H-score, and there admittedly exist other source-free embeddings or metrics (e.g. task2vec, LEEP, LogME) that could also, in principle, be used within it.
> > >
> > > As both an ablation and a further assessment of our framework's flexibility, we have expanded the empirical evaluation on ImageNet-R 10 task benchmark by:
> > >
> > >     1) Replacing H-score with other source-free transferability metrics (LEEP and LogME) while keeping the embedding algorithm and overall pipeline unchanged;
> > >     2) Replacing the entire H-embedding formulation with Task2Vec.
> > >
> > > The results are shown below:
> > >
> > > |HyperNet with H-Embedding (%) | HyperNet with H-embedding-LEEP (%) |  HyperNet with H-embedding-LogME (%) |Hypernet with task2vec (%) |
> > > |-----------|---------|--|--|
> > > | 81.8 | 81.6 |  81.4 | 79.8 |
> > >
> > > As shown, replacing the transferability metric alone does not lead to only minor drop in performance. This supports our view that while H-score is a strong instantiation, the core contribution of our work lies in the transferability-guided embedding formulation and its integration into a hypernetwork-based parameter generation pipeline—a general framework that remains valid even when alternative metrics are employed.
> > >
> > > However, replacing the entire embedding formulation with task2vec leads to a more noticeable performance degradation. We believe this is because Task2Vec computes task embeddings independently without explicitly modeling task-to-task relationships, which limits its capacity to guide transfer dynamics in CL. Additionally, Task2Vec is highly sensitive to the choice and training of the probe network, which introduces extra variance in effectiveness and incurs additional memory costs—challenges not present in our H-embedding approach.
> > >
> > > We will incorporate the above results into the final version along with a more comprehensive discussion. Further exploration of other embedding designs also remains an important direction for our future research.

---

> ### Author Response · Authors · 2025-08-08
>
> **(2) Regarding W2.3 (symmetry and AHP motivation):**
>
> Thank you for raising this point, and we apologize for the misunderstanding in our previous response. Our earlier reply focused on clarifying that aligning the symmetric Euclidean distance to the asymmetric transferability is generally acceptable. Yet, this does not mean that directly aligning them during optimization does not introduce practical issues.
>
> In particular, H-score is a target-centric transferability metric, and its absolute value is largely influenced by the specific target task. This variation misaligns with the symmetry assumption of Euclidean distance: the "score distance" from task A to task B may differ significantly from that from task B to A due to differences in target task property. To mitigate this, we introduced AHP normalization to bring transferability values across tasks into a more consistent range. This normalization helps reduce the scale mismatch and promotes better structural alignment between transferability and Euclidean distance, thus facilitating the optimization of H-embeddings.
>
> We acknowledge that our original explanation (Lines 159–162) may have been unclear and will revise it in the final version to better reflect this motivation.
>
> **(3) Regarding W2.2 (affinity-to-distance conversion):**
>
> Thank you for the further discussion on the affinity-to-distance conversion. We address the points as follows:
>
>  - We are sorry for the lack of clarity in our earlier response. Following the question raised in W2, our previous reply focused on whether reversing the affinity–distance direction was reasonable, without carefully distinguishing the placement of $\gamma$. In our implementation, we actually use $dist = \gamma \cdot \exp(-aff)$. This choice provides a more intuitive scaling in our setting, where the converted distance is then used for alignment. Also, when $\gamma$ is around 1 (as is the case for most of our optimization results), the two forms behave similarly in practice.
>
>  - It appears in caption of Fig. 3 of Taskonomy [Zamir et al., 2018].
>
>  - Thank you for pointing this out. We apologize for the inadequate statement and will revise it for better precision, along with adding necessary discussions in the final version.
>
>  - While we agree that $dist =\exp(-aff)$ can reduce sensitivity to high affinity values, in our setting it is still a reasonable choice, in that: i) H-score and its AHP-normalized version can be sometimes negative, and the exponential ensures positive distances; ii) extreme affinity values may arise from numerical artifacts, so lower sensitivity in this case can be beneficial; iii) within the normal range, the narrowing effect is minor since the derivative of $\exp(-x)$ is -1 near zero. Nevertheless, we acknowledge that there is still room to further investigate the properties of this conversion function, and we thank the reviewer for bringing up this valuable point.
>
> Overall, we recognize that there is room for refinement in the current presentation of our affinity-to-distance conversion, and we will do our best to address these points in the final version to improve rigor. At the same time, we would also like to note that this technical aspect is not central to our framework and does not affect the validity of our main contributions.
>
> **(4) Regarding W2.1 (missing citation):**
>
> The full reference is `Schonfeld, Edgar, et al. "Generalized zero-and few-shot learning via aligned variational autoencoders." Proceedings of the IEEE/CVF conference on computer vision and pattern recognition. 2019.`, where they included a section of related work on *Cross-Reconstruction*. We will ensure that it is properly included in the final bibliography.
>
> We would like to thank the reviewer again for the valuable feedback, which has helped us strengthen the clarity and completeness of the paper. Please let us know if further clarification is needed.

---

> > ### Comment · Reviewer_JmhA · 2025-08-08
> >
> > Thank you to the authors for addressing my main concerns.
> >
> > Regarding the "standard affinity distance," I could not find its definition in the caption of Fig. 3 ("Figure 3: Task Dictionary. Outputs of 24 (of 26) task-specific networks for a query (top left). See results of applying frame-wise on a video here.") in Taskonomy [Zamir et al., 2018]. Are we referring to the same paper?
> >
> > To avoid potential misunderstanding, I suggest changing the phrase "standard affinity distance method" to "affinity distance method used in [Zamir et al., 2018]," or alternatively, using the more intuitive conversion as $dist = \beta (1 - aff)$.
> >
> > Since my main concerns have been addressed, I will raise my score.

---

> > > ### Author Response · Authors · 2025-08-08
> > >
> > > Thanks for your constructive follow-up and for your willingness to raise the score. We sincerely appreciate your careful reading of our work. We apologize for the typo in our earlier response — the correct reference should be Figure 7 (rather than Figure 3) in Taskonomy [Zamir et al., 2018]. Regarding the wording issue, we will adopt your suggestion to revise it in the final version, as well as try the alternative conversion you proposed. Thank you again for your helpful feedback and for your kind consideration of a higher score for our work.

---

### Note · Authors · 2025-08-12

We sincerely thank the AC and reviewers for their time, thoughtful feedback, and insightful suggestions throughout the review process, which have greatly helped us clarify and improve our work.

Our paper addresses an important gap in CL by introducing prior information—derived from transferability—about task relationships into CL procedures. Unlike most existing CL methods that focus on task models, our key insight is that task relationships encode valuable information about the task space structure in CL, and its utilization can notably enhance both forward and backward transfer. Specifically, we propose a transferability-aware task embedding (H-embedding) and construct a hypernet framework under its guidance to learn task-conditioned model weights for CL tasks. The embedding is designed to be online and easy to compute, fully adhering to rehearsal-free CL constraints. Meanwhile, the framework is lightweight, efficient, and compatible with various backbones and PEFT techniques such as LoRA, making it practical and ready-to-use with pretrained models. The effectiveness of our framework is supported by extensive experiments across different backbones and CL benchmarks.

In the rebuttal process, we carefully addressed all raised concerns, including clarifying our module and formula implementations and their motivations, explaining experimental results, refining ambiguous wording, and adding discussions and new experiments on technical aspects such as hyperparameter sensitivity, alternative metric choices, and efficiency in training and inference. We believe these efforts reinforce the soundness and rigor of our approach. During the discussion, all reviewers indicated that their main concerns had been resolved, and they either increased their scores or maintained their positive evaluations accordingly.

We have also acknowledged areas for further refinement in the final version. These planned improvements primarily focus on enhancing clarity, rigor, and the breadth of discussion, while the core ideas and conclusions of our work remain intact. In the final version, we will integrate these clarified presentations and expanded discussions to further strengthen the paper.

Overall, we believe the clarifications and additional analyses have improved the presentation and reinforced the validity of our contributions. We greatly appreciate the opportunity for this constructive discussion and look forward to the AC’s assessment in light of our responses and updates.

---

### Decision · Program_Chairs · 2025-09-17

**Decision:**

Accept (poster)

**Comment:**

This paper extends the continual learning with hypernetworks (CLH) framework by introducing an additional objective that helps model the relationships between tasks, compared to simply learning task embeddings end-to-end as in the original CLH paper. The experiments are conducted on an array of well-chosen task families, and show some performance improvements over the original CLH method.

The reviewers generally found the paper to be well-written and appreciated its strong set of experiments. While the improvements over vanilla CLH may not appear to be very impressive, they are still significant, and CLH is a strong baseline method for the continual learning problems considered by the authors. I'm voting for acceptance.